# Associations between fMRI signal amplitude, hemispheric asymmetry, and task performance
Dardo Tomasi ✉ & Nora D. Volkow

Interhemispheric asymmetry is a core feature of human brain organization, yet its functional relevance across cognitive tasks remains incompletely understood. Using data from 989 healthy adults, we examined patterns of functional asymmetry and their relationship to bilateral fMRI signal amplitude and task performance across seven tasks: motor, language, social cognition, relational processing, working memory, gambling, and emotion. An fMRI-derived asymmetry index was computed across 17 task epochs and mapped onto the cortical surface. Here we show that both fMRI signal amplitude and asymmetry were positively associated with task accuracy across multiple networks and tasks epochs. These associations were strongest in language, frontoparietal, and dorsal attention networks during high-demand tasks, such as story comprehension, relational processing, and working memory. Partial least squares regression revealed that amplitude was a more robust predictor of task accuracy than asymmetry. These findings suggest that greater neural activation drives stronger hemispheric differentiation and supports cognitive performance.

Interhemispheric differences in brain function, also known as functional brain asymmetry, are of long-standing interest in neuroscience due to their critical role in language, attention, and motor control[1–8]. The left hemisphere is typically dominant for language[9], and relational processing, which relies on the left anterior prefrontal cortex[10,11], whereas the right hemisphere more often supports spatial attention, facial recognition, and emotion processing[12,13]. Evidence suggests that individuals with greater task-related asymmetry may have higher accuracy, consistent with the notion that lateralized activation patterns reflect the optimized recruitment of specialized cortical systems[14]. Supporting this, a recent large-scale meta-analysis found that regions exhibiting greater lateralization tend to have weaker interhemispheric structural connectivity[15], implying that such regions rely less on costly interhemispheric integration and instead favor efficient unilateral processing during demanding tasks.

The lateralized functions of the hemispheres have clear clinical implications. Lesions to the left hemisphere often result in language impairments (aphasia), whereas right hemisphere damage more frequently leads to alterations in emotional and social behavior[16,17]. Psychiatric conditions also involve disruptions in hemispheric balance; for example, depression is associated with reduced left-hemisphere activity related to positive affect and reward processing and increased right-hemisphere activity linked to negative emotion[18]. Similar alterations have been observed in schizophrenia[19–21] and in normal aging[22,23]. These findings underscore the relevance of functional asymmetry to understanding brain-behavior

relationships in health and disease. However, the literature on functional asymmetry yields highly variable findings, complicating systematic comparisons of asymmetry indices across studies[24].

Despite long-standing interest, the behavioral significance of individual differences in hemispheric specialization remains unresolved. While greater lateralization has been linked to more efficient neural processing in healthy individuals[25], reductions in asymmetry have been interpreted in conflicting ways, either as adaptive compensatory recruitment[22]; or as 'dedifferentiation' reflecting inefficient processing due to failure to suppress task-irrelevant activity[26]. Furthermore, it is still unclear how deviations from typical patterns impact task performance[27].

Surface-based analyses offer a precise approach for studying brain asymmetry, particularly because they focus on the cortical sheet where most higher-order cognitive functions occur[28]. The Human Connectome Project (HCP) (https://www.humanconnectome.org/), a large-scale publicly available imaging dataset, provides high-quality, standardized fMRI data in surface space. This dataset offers a unique opportunity to evaluate how task-evoked asymmetry relates to behavioral performance using consistent acquisition parameters, robust preprocessing, and well-characterized tasks. Leveraging the HCP task-fMRI data, our study aims to clarify the functional significance of hemispheric asymmetry by examining how lateralized brain activity varies across fMRI tasks and how it relates to task performance. Using a threshold-independent approach to compute asymmetry, we test whether greater lateralization is associated with higher task accuracy,

National Institute on Alcohol Abuse and Alcoholism, Bethesda, MD, USA. ✉e-mail: dardo.tomasi@nih.gov

particularly under greater cognitive demand, in line with theories of neural efficiency and specialization.

In line with prior reports that greater hemispheric lateralization in task-evoked activity is often accompanied by fMRI signal[29], we hypothesized a positive linear association between fMRI signal amplitude and hemispheric asymmetry, and that task accuracy would increase with both measures, particularly in cognitively demanding tasks. Although canonical lateralization profiles have been described for language, motor, emotional, and other domains, it remains unclear how reproducibly these patterns emerge within individuals across separate datasets. To address this gap, we hypothesized that distinct task epochs would yield consistent asymmetry profiles —such as left-hemisphere dominance for language and relational processing, contralateral motor patterns, and right-hemisphere dominance for emotion and social cognition—and tested this in two matched HCP subsamples, enabling a direct, cross-cohort reproducibility assessment. Asymmetry was expected to be most pronounced in the somatomotor (SMM) and language (LAN) networks.

## Results

We utilized fMRI datasets from the publicly available WU-Minn HCP 1200 Subjects data release (http://www.humanconnectome.org/). A total of 989 participants, each of whom completed seven fMRI tasks (motor, language, social cognition, relational processing, working memory, gambling, and emotion) with root-mean-square (RMS) head displacement<2 mm, were included (Fig. 1a).

We analyzed 16,853 individual fMRI signal contrasts corresponding to 17 conditions per participant, involving 91,281 grayordinates within the brain[30]. A functional asymmetry index, $\Delta$, the normalized difference between left (L) and right (R) hemisphere fMRI signal amplitudes[28],

$$\Delta = \frac{L - R}{|L| + |R|},\tag{1}$$

and the mean bilateral amplitude of the fMRI signal,

$$A = \frac{|L| + |R|}{2},\tag{2}$$

were mapped to the 32,492 vertices of the left cortical hemisphere for each contrast independently (Fig. 1b). To evaluate the reproducibility of the asymmetry index, we divided the participants into Discovery ($n = 504$) and Replication ($n = 485$) subsamples, while matching these cohorts by age, sex, and body mass index (BMI) using the caTools R-library (https://cran.r-project.org/web/packages/caTools/). There were no significant differences in sex, age, race, BMI, handedness, and task accuracy between Discovery and Replication subsamples (Supplementary Table S1).

### Vertex-wise interhemispheric asymmetry

To comprehensively characterize functional asymmetry across sensorimotor and higher-order domains, we included all available HCP tasks and contrasts in our analysis. This approach allowed us to benchmark well-established motor asymmetries and systematically compare the strength and reproducibility of lateralization across brain systems and task types within the same dataset.

fMRI signal asymmetry varied across tasks (Fig. 2 and Supplementary Fig. S1; $P < 0.05$, FDR-corrected). In the motor task, a one-sample $t$ test

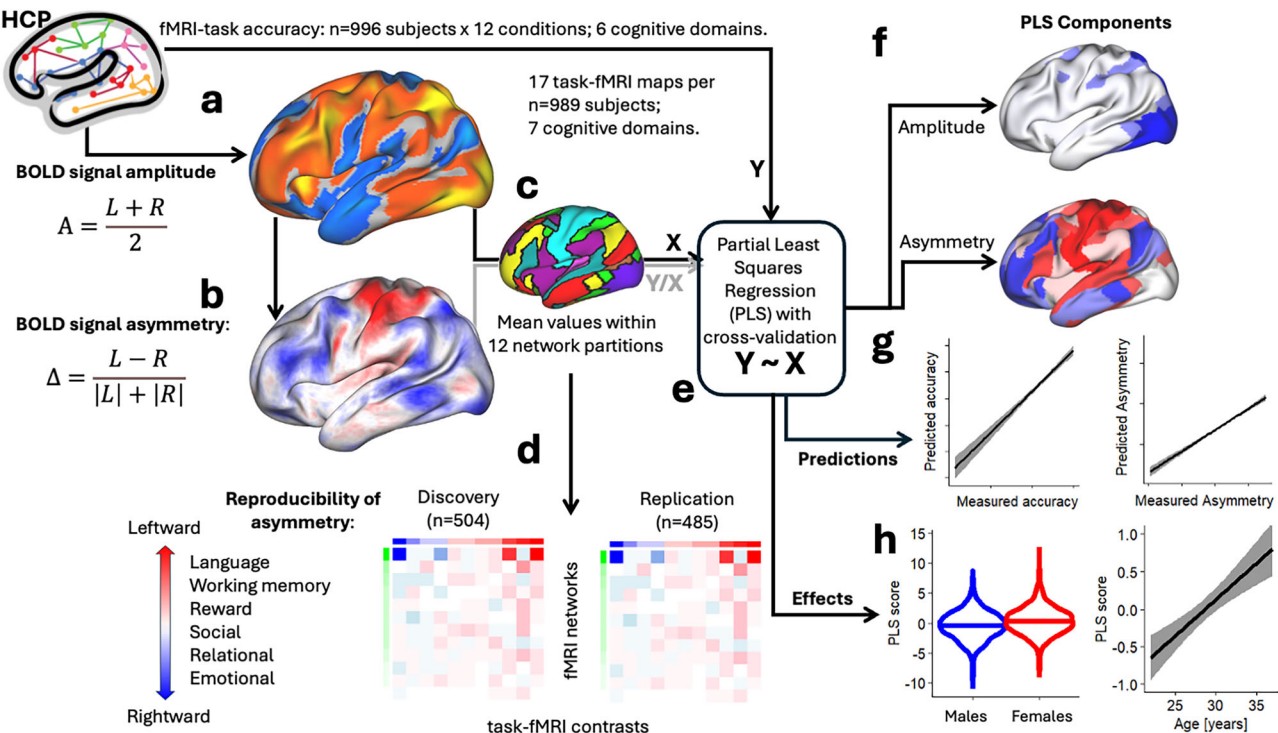

**Fig. 1 | Study flow chart.** fMRI contrasts across 17 task epochs were analyzed for 989 healthy young adults from the Human Connectome Project (HCP; **a**). Left (L) and right (R) hemisphere fMRI signals were assessed independently at 32,492 cortical vertices per participant and contrast to quantify functional interhemispheric asymmetry, $\Delta$ using Eq [1] (**b**). Individual maps of fMRI signal amplitude and asymmetry were averaged across 12 predefined network partitions[31] for each participant and contrast (**c**). The sample was divided into Discovery ($n = 504$) and Replication ($n = 485$) subsamples to examine the reproducibility of the functional asymmetry index. Functional networks were ranked by root-mean-square values of asymmetry, and task contrasts by mean asymmetry values across participants within each subsample (**d**). Partial least squares (PLS) regression (**e**) was applied to extract principal components of fMRI signal amplitude and asymmetry (**f**). These components were used to predict asymmetry from fMRI signal amplitude, and task accuracy independently for amplitude and asymmetry (**g**), and assessed the effects of age and sex on PLS regression scores from amplitude and asymmetry (**h**).

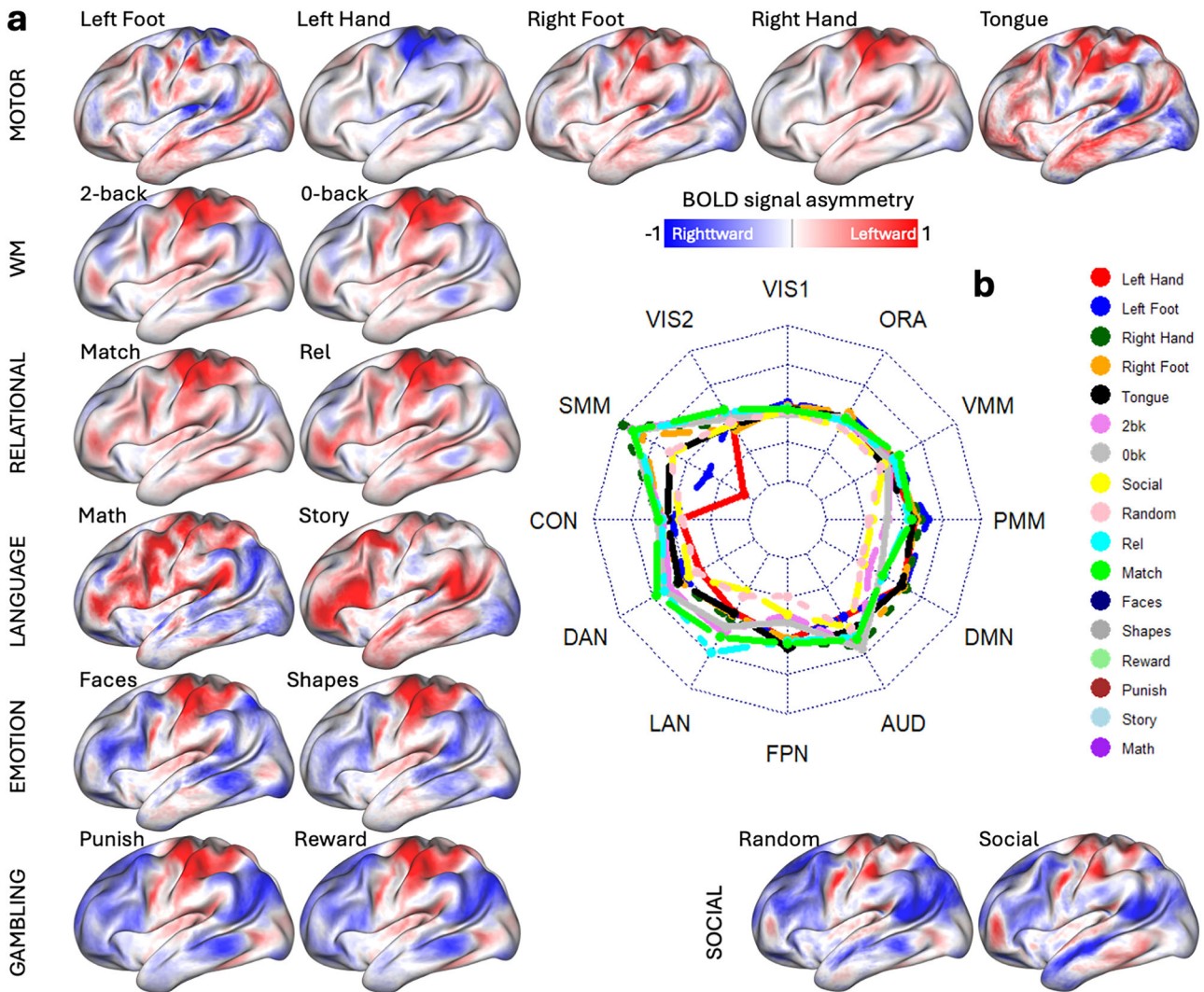

**Fig. 2 | Functional asymmetry maps.** Vertex-wise asymmetry maps for 17 epochs of 7 fMRI tasks rendered on the left cortical hemisphere (**a**). The sample includes 989 healthy young adults. One-sample *t* test. *P* < 0.05, FDR-corrected for multiple comparisons. The radar chart (**b**) summarizes average asymmetry for each task epoch across 12 network partitions: Visual (VIS1 and VIS2), Somatomotor (SMM), Cingulo-Opercular (CON), Dorsal Attention (DAN), Language (LAN), Fronto-parietal (FPN), Auditory (AUD), Default-Mode (DMN), Posterior (PMM), Ventral Multimodal (VMM), and Orbito-Affective (ORA) networks. Each spoke represents one of the 12 functional networks. For each network, the radial distance from the center reflects the average magnitude of asymmetry across all 17 task contrasts. Positive values (outward) indicate leftward asymmetry, and negative values (inward) indicate rightward asymmetry. Radar chart inner and outer limits are set at −0.3 and 0.3, respectively.

revealed opposing interhemispheric asymmetry, primarily in the SMM cortex, a partition of the Cole-Anticevic brain network atlas[31]. Specifically, left-hand/foot movements showed greater activation in the right hemisphere, while right-hand/foot movements activated the left hemisphere more. For all tasks, except those involving left limb movements and possibly language, leftward asymmetry in the sensorimotor cortex likely reflects activation related to the contralateral (right) response limb (Fig. 2). The radar plot in Fig. 2b summarizes the asymmetry index values across functional networks for each task epoch. Each axis corresponds to a functional network, and the distance from the center reflects the magnitude of average asymmetry, with concentric circles representing increasing values (e.g., −0.30, 0.15, 0.00, 0.15, 0.30) that indicate greater leftward lateralization.

To isolate language-specific processes while controlling for auditory, attentional, and working memory demands, we used the story > math contrast that has been validated in prior work as a reliable marker of language-related lateralization[32]. For the language task, story listening, compared to solving auditory arithmetic problems, showed greater leftward asymmetry in the LAN, frontoparietal (FPN), and default mode networks (DMN), with rightward asymmetry in the visual and SMM cortex (Fig. S1).

A similar asymmetry pattern appeared when contrasting relational (relation vs. match epochs) or social (social vs. random) task epochs (Fig. S1). Higher working memory load (2-back>0-back) was associated with leftward asymmetry in visual and SMM cortex and rightward asymmetry in DMN regions (Fig. S1).

In the emotion task, matching faces versus shapes was linked to pre-dominant rightward asymmetry in the dorsal attention (DAN), FPN, and posterior multimodal (PMM) networks (Fig. S1). For the gambling task, asymmetry differences between reward and punishment epochs were minimal (Fig. S1). The asymmetry patterns are reproduced in Discovery and Replication subsamples (Fig. S2 and Supplementary Statistical results in CIFTI format).

## Networks and tasks ranked by asymmetry
We used established functional network partitions to organize regional fMRI signals, enabling a structured analysis of hemispheric asymmetry across the brain. The DMN was included because, despite being typically deactivated during tasks, it plays a crucial role in internally directed cognition and interacts with task-positive networks, potentially influencing

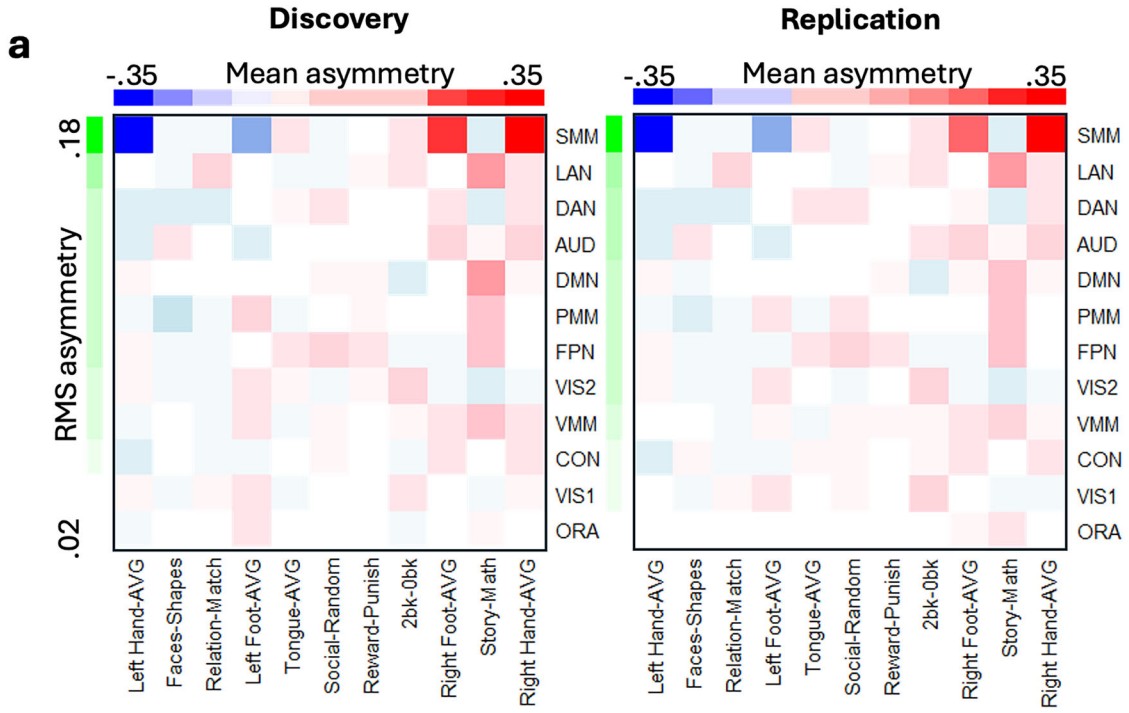

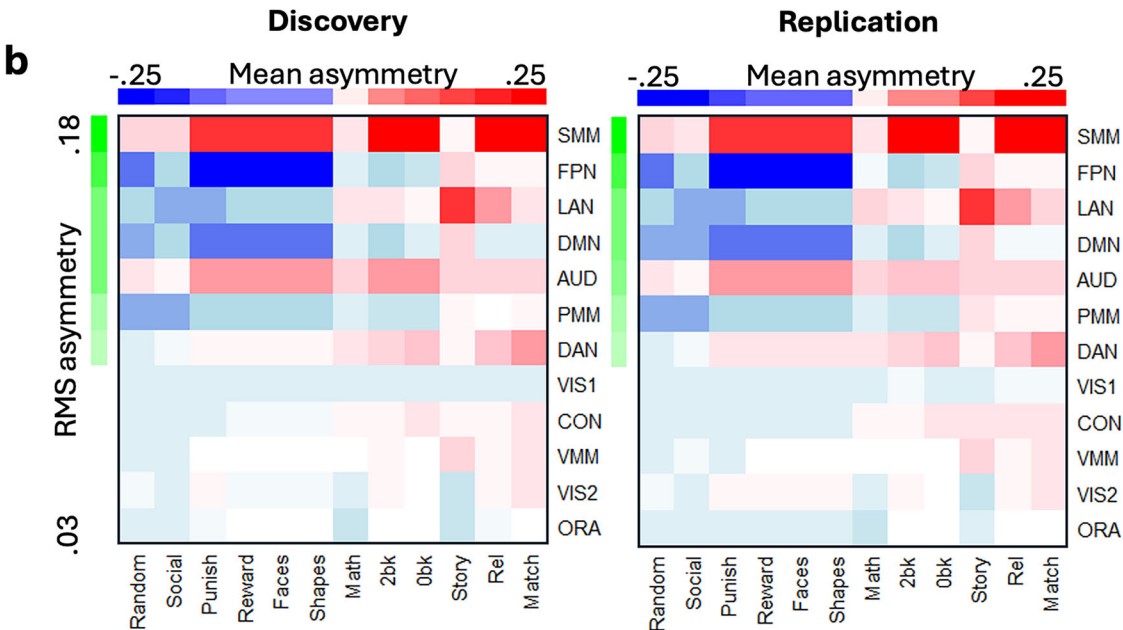

**Fig. 3 | Network asymmetry.** Average asymmetry within nine functional networks is ranked by root-mean square (RMS) values and by mean asymmetry across 11 task contrasts (**a**) or 12 task epochs (**b**), independently for the Discovery (*n* = 504) and Replication (*n* = 485) subsamples. **a** each motor condition (left or right hand, foot, or tongue) was contrasted against the average (AVG) activation across all motor conditions. Network abbreviations: VIS1 and VIS2 visual, CON cingulo-opercular DAN dorsal-attention, SMM somatomotor, FPN frontoparietal, AUD auditory, DMN default mode, PMM and VMM Posterior and Ventral Multimodal, ORA orbito-affective.

asymmetry patterns. Specifically, we averaged asymmetry maps within 12 canonical network partitions[31], for each participant and fMRI contrast (Fig. 1c). This network-based approach provides a systems-level perspective that complements traditional anatomical labels by capturing functionally coherent regions known to support distinct functional domains. To characterize asymmetry by network, we ranked networks based on the RMS of asymmetry across contrasts. Conversely, to evaluate asymmetry by contrast, we ranked contrasts by their mean asymmetry across networks (Fig. 1d).

Across task contrasts, interhemispheric asymmetry was most pronounced in the SMM network (Fig. 3a, b; $P < 0.05$, FDR-corrected), revealing a reproducible yet complex pattern. For example, relative to the overall asymmetry induced by the motor task, movements of the left hand/foot were linked with rightward asymmetry in SMM, DAN, and auditory (AUD) networks, while movements of the right hand/foot showed the opposite pattern (Fig. 3a). In the language task, compared to math epochs, story epochs showed rightward asymmetry in SMM, DAN, and visual (VIS1

and VIS2) networks, and leftward asymmetry in LAN, AUD, DMN, PMM, FPN, and ventral multimodal (VMM) networks (Fig. 3a).

Across all task epochs, the SMM and AUD networks exhibited consistent leftward asymmetry except for rightward asymmetry with movements of left hand/foot and the DMN exhibited consistent rightward asymmetry, except for the story epoch of the language task (Fig. 3b). Differently, FPN and LAN networks displayed notable rightward asymmetry during social (random and social), gambling (punish and reward), emotion (faces and shapes), and working memory (0-back and 2-back) task epochs, with a marked leftward asymmetry during the story epochs of the language task (Fig. 3b).

## Association between fMRI signal amplitude and asymmetry

Because the asymmetry index is derived from the difference in fMRI signal amplitude between corresponding regions in the left and right hemispheres, it is inherently influenced by the magnitude of task-evoked activation. Regions with stronger activation are more likely to exhibit detectable asymmetries, whereas low-activation regions may yield smaller or noisier asymmetry estimates. To identify spatial patterns linking mean bilateral amplitude (Eq [2]) to asymmetry (Eq [1]) across the cortex, we performed vertex-wise correlations between these two measures and averaged the resulting maps across all 17 task epochs. Averaging across tasks reduces variability that arises from task-specific activation differences, thereby highlighting consistent, task-general associations between amplitude and asymmetry.

The strongest positive associations ($R > 0.65$; corrected Cohen's $d > 1.7$) were observed in lateral occipital, ventral temporal, inferior frontal, and inferior parietal regions (Fig. 4a). In these areas, low average activation was associated with strong rightward asymmetry, which decreased as activation increased (Fig. 4b), indicating that greater neural engagement reduces rightward lateralization. In contrast, weaker correlations ($R < 0.55$, $P < 2.2e{-}16$; $1.15 < d < 1.3$) were observed in medial prefrontal, superior frontal, and medial parietal cortices, including regions of the DMN, where asymmetry was generally low and its relationship with amplitude correspondingly modest. These results indicate that the amplitude–asymmetry coupling is strongest in regions that exhibit robust lateralization, such as those supporting language and executive functions, and diminishes in areas with minimal intrinsic asymmetry.

To examine amplitude–asymmetry relationships at the network level, we computed, for each participant, the mean fMRI signal amplitude within each of nine major networks, separately for the left and right hemispheres and each of the 17 task epochs (Fig. S1; Supplementary Statistical Results in CIFTI format). Asymmetry and amplitude were then averaged independently per network and epoch, and correlations were computed across participants. This approach complements the vertex-wise analysis by reducing spatial variability and emphasizing consistent network-level effects. Using this method, the strongest amplitude–asymmetry association was observed in the LAN network during the Random epoch (Fig. 4c), consistent with the vertex-wise findings and supporting a robust coupling between neural engagement and lateralization in language-related regions.

## fMRI signal amplitude predicts asymmetry

We applied partial least squares (PLS) regression to investigate whether the mean interhemispheric fMRI signal amplitude predicts asymmetry (Fig. 1e, g). PLS is designed to handle scenarios where predictors (e.g., mean amplitudes) are highly correlated across networks and contrasts (multicollinear)[33] as in this study (Fig. S4). Different from multiple linear regression, which can lead to unstable or overfitted models, PLS reduces the data's dimensionality by constructing orthogonal components that maximize the covariance between predictors and the response variable (e.g., asymmetry)[34]. A total of 153 predictors (mean fMRI signal amplitude values within 9 networks for 17 task epochs) were used to estimate 153 response variables (mean asymmetry values within 9 networks for 17 task epochs) using 12 PLS components with within-sample 10-fold cross-validation.

These components captured 63% of the variance, providing robust predictions without overfitting.

To evaluate the robustness of PLS predictions in new samples, we trained the PLS model using within-sample 10-fold cross-validation in the Discovery sample, then applied this optimal model to predict asymmetry from amplitudes in the Replication sample (Fig. S5a, b). To complete the 2-fold cross-validation, we reversed the process: training the PLS model in the Replication sample and using it to predict asymmetry from amplitude in the Discovery sample (Fig. S5c, d). The 2-fold cross-validation across test subsamples was successful, reproducing the same prediction patterns observed in the training.

PLS regression revealed a complex predictive pattern. For epochs within the social and gambling tasks, the amplitude regressors successfully predicted asymmetry primarily in the LAN, DMN, FPN, and cingulum-opercular (CON) networks, with lesser contributions from VIS1, VIS2, and DAN (Fig. 4c; $P < 0.05$, Bonferroni-corrected). Amplitude regressors also predicted asymmetry in VIS1 and VIS2 for working memory, relational, and emotion task epochs, as well as in VIS2 for motor task epochs ($P < 0.05$, Bonferroni-corrected). Further predictions were seen in DMN and FPN for the math epoch ($P < 0.05$, Bonferroni-corrected).

## Task accuracy

Participants demonstrated high overall accuracy across the task-fMRI paradigms (Fig. 5a and Supplementary Table S1). Accuracy was calculated for each individual and task based on correct responses during scanner-based performance. For the emotion, language, relational, and working memory tasks, the mean accuracy was above 60%, indicating good task engagement and compliance. In contrast, average accuracy fell to chance level (~50%) for both the gambling and social cognition tasks, reflecting either the probabilistic nature of task outcomes or limitations in performance measurement in those paradigms. Note that the motor task was designed primarily to map robust sensorimotor activation rather than assess performance variability, and therefore does not include accuracy metrics, as the simple, cued movements are expected to elicit high compliance with minimal behavioral variance.

We next assessed whether task accuracy was associated with the amplitude and asymmetry of fMRI responses across fMRI tasks. Greater signal amplitude and asymmetry (Fig. 5b–e) were significantly correlated with better accuracy in multiple networks and task epochs ($P < 0.05$, Bonferroni-corrected). Notably, these effects were most consistent in LAN, VIS1, VIS2, DAN, and FPN, particularly during relational processing, working memory (2-back and 0-back), and story comprehension epochs. For instance, in the LAN during the story epoch, both amplitude and asymmetry were positively associated with task accuracy (Fig. 5f–h), with asymmetry explaining a slightly larger proportion of variance ($R^2 = 0.17$) than amplitude ($R^2 = 0.13$). These findings support the idea that increased engagement and lateralization of functional activity are beneficial for cognitive performance, particularly in tasks requiring language processing, attention, and memory. Higher-order polynomial models (up to the third degree) linking asymmetry to amplitude, and task accuracy to either amplitude or asymmetry did not improve model fit significantly over simpler linear models ($\Delta AIC < 3.7\%$; Fig. 5f–h), suggesting that the associations between these neural measures and behavioral performance are predominantly linear in nature.

## fMRI signal amplitude predicts task accuracy better than asymmetry

We also applied PLS regression, using within-sample 10-fold cross-validation, to predict task accuracy from either mean fMRI signal amplitude or interhemispheric asymmetry (Fig. 1e, g). Specifically, we used 153 predictors (mean amplitudes or asymmetry values within 9 major networks for 17 task epochs) to predict task accuracy across 12 task epochs: emotion (faces, shapes), gambling (reward, punish), language (story, math), relational processing (rel, match), social cognition (social, random), and working memory (0-back, 2-back). Prediction robustness was further validated using

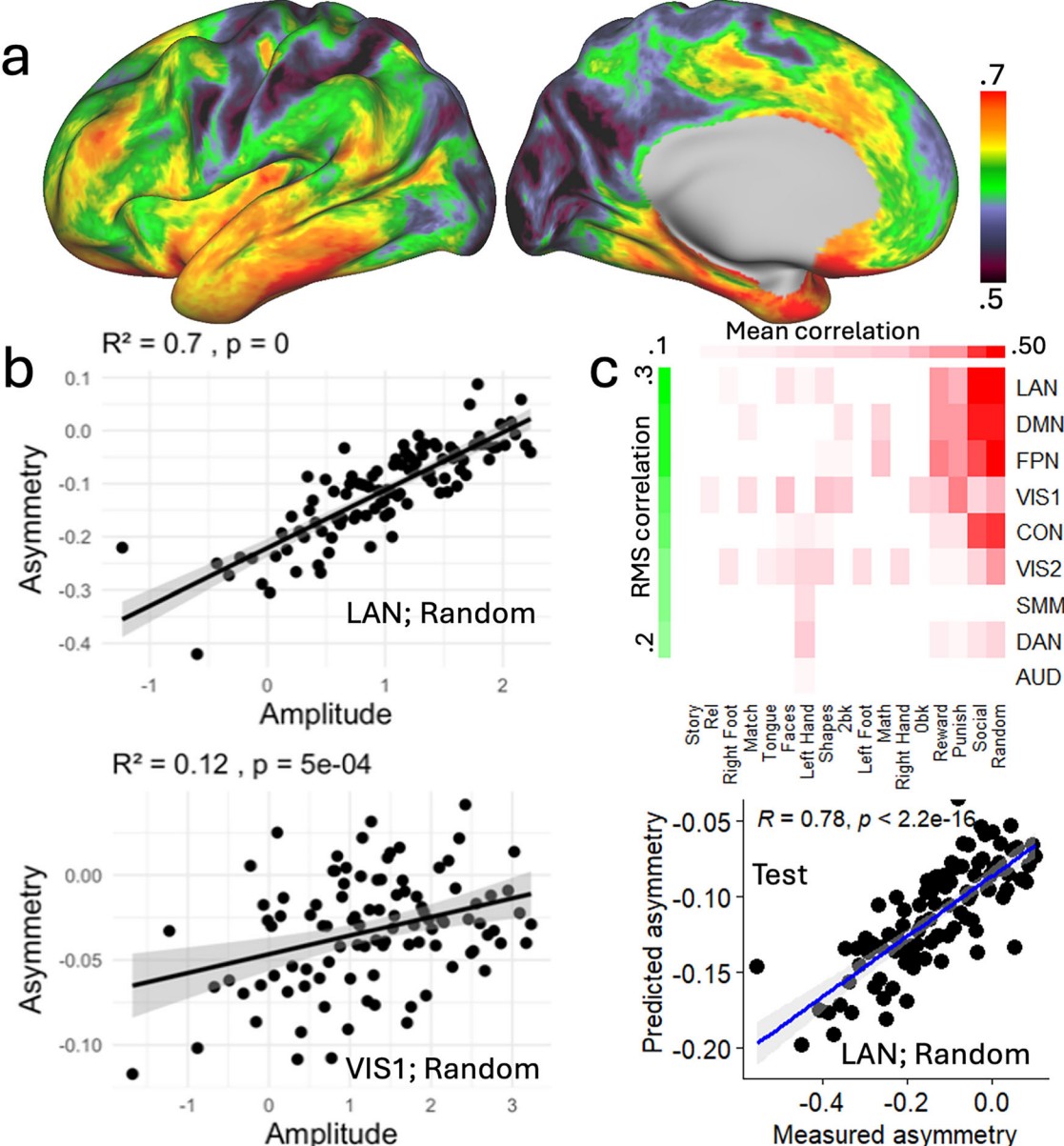

**Fig. 4 | Linear association between fMRI signal asymmetry and amplitude.**
**a** Average correlation between hemispheric asymmetry and BOLD signal amplitude across 17 task epochs, rendered on the lateral (left) and medial (right) surfaces of the left cortical hemisphere ($0.25 < R^2 < 0.5$; $P < 2.^2$e-16). **b** Representative scatter plots illustrating the linear relationship between asymmetry and amplitude in the language (top) and primary visual (bottom) networks during the random epoch of the social cognition task, which was the task epoch that showed the strongest association. Each point reflects the average of 10 individuals grouped by similar fMRI amplitude. **c** Partial least squares (PLS) prediction of asymmetry based on mean bilateral BOLD amplitude across 17 task contrasts and 9 major networks. The top matrix shows correlations between predicted and observed asymmetry, ranked by mean values across epochs (columns) and by root-mean-square values across networks (rows), thresholded at $P < 0.05$ (uncorrected). The bottom scatter plot highlights prediction performance for the language network during the random epoch. For display purposes, individuals were ranked based on their average bilateral amplitude to form 100 groups of 10 individuals with similar amplitude levels, independently for each task epoch and network. Then, for each group, task epoch, and network, we computed the average amplitude and average asymmetry to assess the relationship between these metrics at a group level. Network abbreviations: VIS1 and VIS2 visual, CON cingulo-opercular, DAN dorsal attention, SMM somato-motor, FPN frontoparietal, AUD auditory, DMN default mode. Sample: 989 healthy adults.

twofold cross-validation in the Discovery and Replication subsamples, as for asymmetry predictions from fMRI amplitudes.

As shown in Fig. 6a, both amplitude- and asymmetry-based models produced robust predictions of task accuracy, with consistent results observed in both Discovery and Replication samples. Prediction accuracy was highest for the 2-back, match, and the story epochs, particularly when using amplitude-based predictors. Although amplitude consistently outperformed asymmetry overall, asymmetry still accounted for significant variance in task accuracy across most conditions. The heatmaps also demonstrate strong generalizability, with prediction performance maintained from training to test data and across datasets. This is further illustrated in the scatter plots for the 2-back task (Fig. 6b), where amplitude-based predictions yielded a very strong linear association with measured accuracy ($p < 2.2 \times 10^{-16}$), and asymmetry-based predictions were also significant ($p = 3.2 \times 10^{-6}$), albeit with lower explained variance.

### Spatial components
PLS regression revealed spatially distributed patterns across brain networks, with each component capturing distinct regions where either fMRI signal

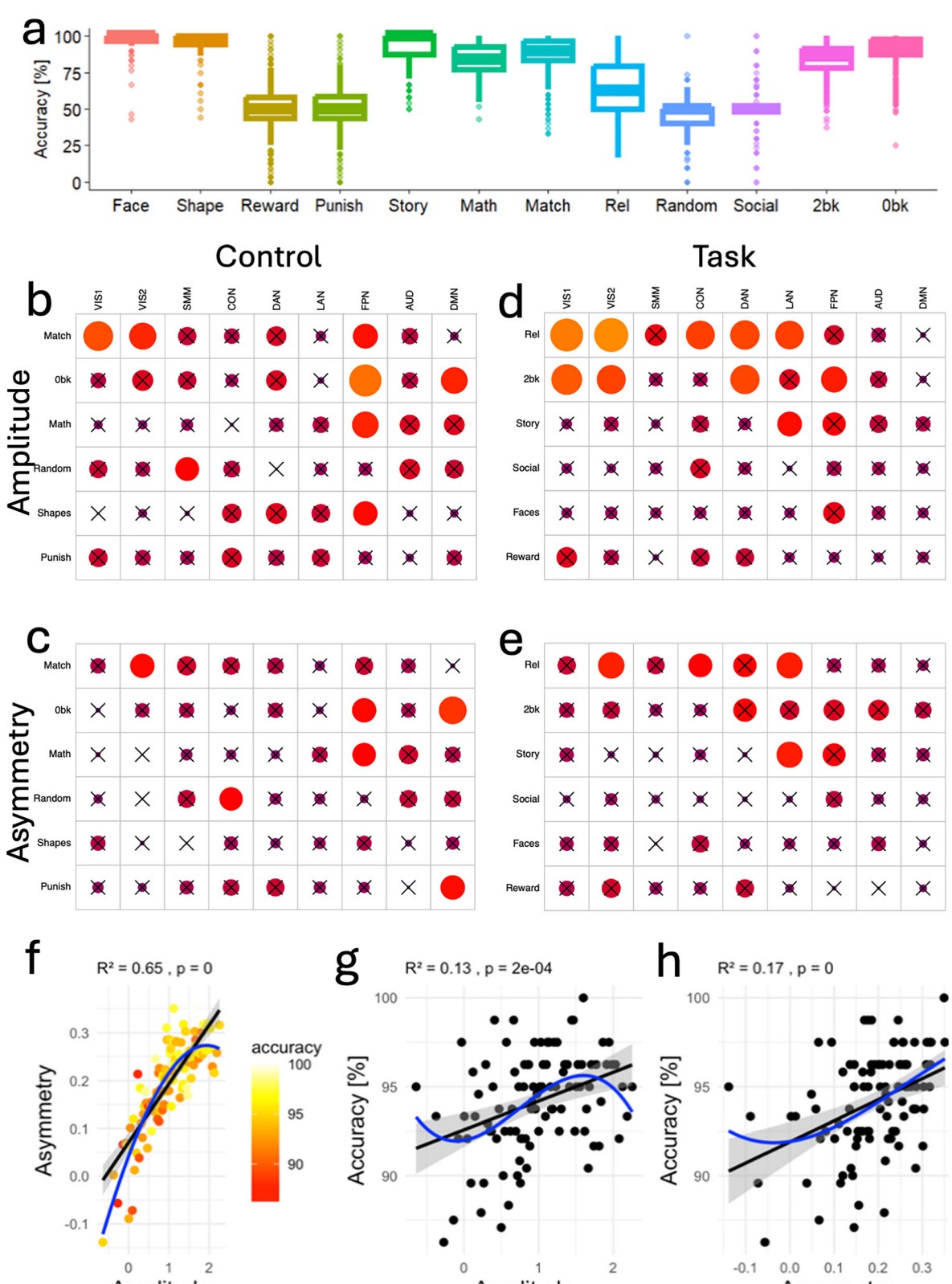

amplitude or asymmetry contributed to task-related variance in accuracy. The 12 primary components of fMRI signal amplitude captured 57% of the variance in the Discovery sample and 59% in the Replication sample. These components showed robust loadings in specialized networks, such as VIS1, VIS2, and DAN (components 1 and 2; Fig. 7). Additional components highlighted functional contributions from broader networks, including

DMN and FPN, aligning with task demands across higher-order processing tasks. Spatial maps for each component displayed consistent patterns across cross-validation folds, confirming that these PLS-derived networks are reproducible and stable predictors of task-relevant neural responses. For asymmetry, the 12 primary components captured 46% of the variance in the Discovery sample and 45% in the Replication sample, with prominent

**Fig. 5 | Task accuracy vs signal amplitude and asymmetry. a** Boxplot representing the distribution of performance accuracy across 12 main task epochs. The box spans the interquartile range (IQR), containing the middle 50% of data points, and the line within each box denotes the median. Whiskers extend from each box to the smallest and largest values within 1.5 times the IQR from the first and third quartiles. Data points beyond the whiskers are displayed as outliers, representing values outside this range. **b–e** Pearson correlations between accuracy and fMRI signal amplitude (**b** and **d**) or asymmetry (**c, e**) for nine major networks and for control (**b, c**) and task (**d, e**) epochs. Non-significant correlations ($p \geq 0.001$) are marked with a black cross. The color of each circle represents the direction and strength of the correlation, while the area of each circle is proportional to the value of the correlation coefficient, reflecting its magnitude. **f–h** Representative scatter plots illustrating linear relationships between accuracy (color coded in **f**) and both activation amplitude (**g**) and hemispheric asymmetry (**h**) within the language network during the story epoch of the language task. Each point reflects the average of 10 individuals grouped by similar amplitude. Black and blue lines are 1st and 3rd order polynomial fits to the data. The shaded area represents the 95% confidence interval around the fitted line. Network abbreviations: VIS1 and VIS2 visual, CON cingulo-opercular, DAN dorsal attention, SMM somatomotor, FPN frontoparietal, AUD auditory, DMN default mode. Sample: 989 healthy adults.

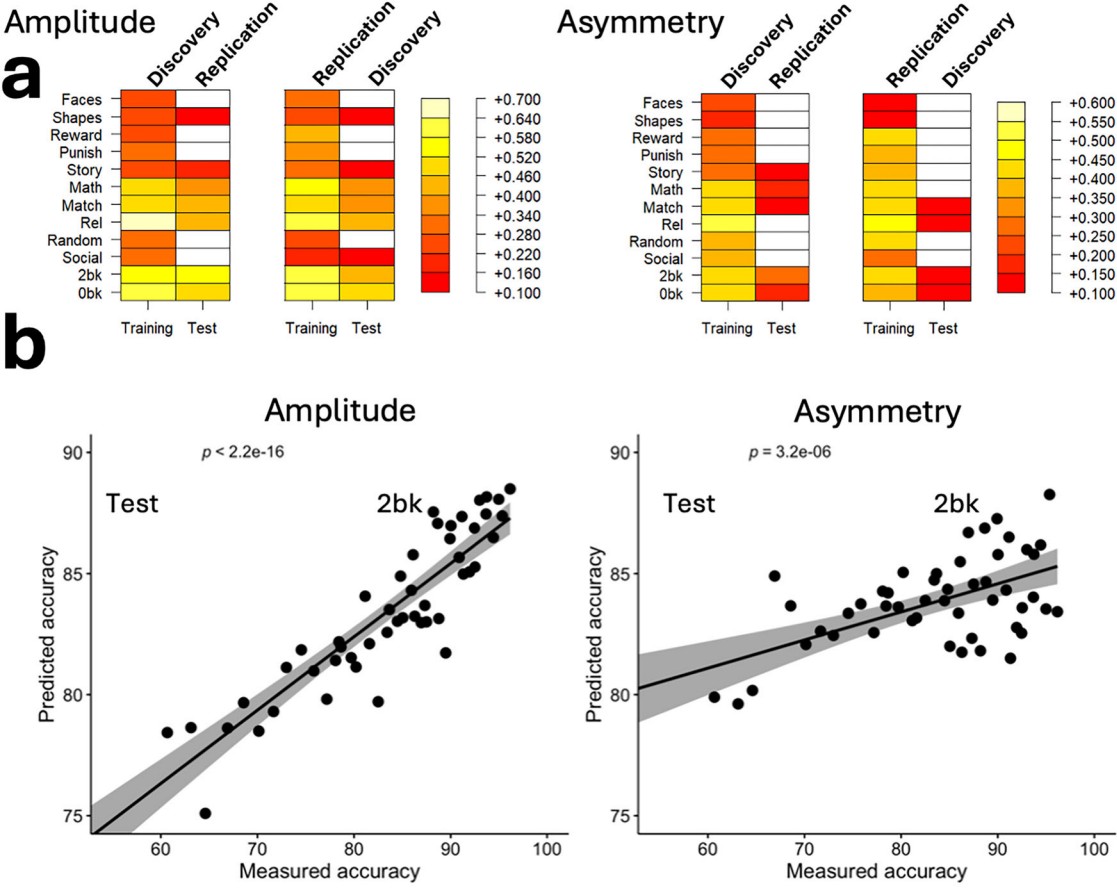

**Fig. 6 | Prediction of task accuracy. a** Significant correlations between measured accuracy and accuracy predicted from fMRI signal amplitude (left) and fMRI signal asymmetry (right) in the Training and Test samples, thresholded at $P < 0.05$ (Bonferroni-corrected for multiple comparisons). Predictions were derived using partial least squares (PLS) regression based on average fMRI measures within nine major networks across 17 task epochs. **b** Representative scatter plots illustrating the linear association between measured and predicted accuracy in the Replication Test sample, shown for the 2-back epoch of the working memory task. Data are shown for Discovery ($n = 504$) and Replication ($n = 485$) subsamples.

loadings in LAN and SMM. A combined PLS model incorporating both amplitude and asymmetry metrics did not outperform the amplitude-only model. This suggests that although asymmetry reflects distinct spatial features associated with task performance, its predictive contribution is secondary to that of amplitude.

## Effects of age and sex

PLS analysis of accuracy predictions based on fMRI signal amplitudes revealed significant effects of age and sex on component scores. Specifically, scores for component 2, which prominently loaded on VIS1, VIS2, and DAN networks, were higher in male compared to female participants and showed an age-related decrease across both Discovery and Replication samples ($P < 0.003$; Fig. S6). For accuracy predictions based on asymmetry, PLS scores for components 1 and 3, which were heavily loaded on LAN and

SMM networks, respectively, were significantly higher in female participants than in males ($P < 0.03$; Fig. S7).

## Discussion

Here, we examined the role of functional asymmetry in supporting task-specific processing across a range of fMRI tasks with a laterality index applied directly to the cortical sheet. This approach enabled precise, vertex-wise measurement of lateralization across diverse functional networks, enhancing our understanding of the spatial patterns of brain asymmetry. Consistent with previous findings[9,12,13], we observed distinct lateralization patterns associated with language, emotional processing, and motor control tasks, corroborating long-standing theories of hemispheric specialization[35]. Notably, all findings were highly reproducible across Discovery and Replication samples, underscoring the robustness of these lateralization

**Fig. 7 | PLS components.** The first six spatial components derived from partial least squares (PLS) regression models based on fMRI signal amplitude (left) or asymmetry (right), with corresponding explained variance (%) indicated. Results are from the discovery subsample (*n* = 504).

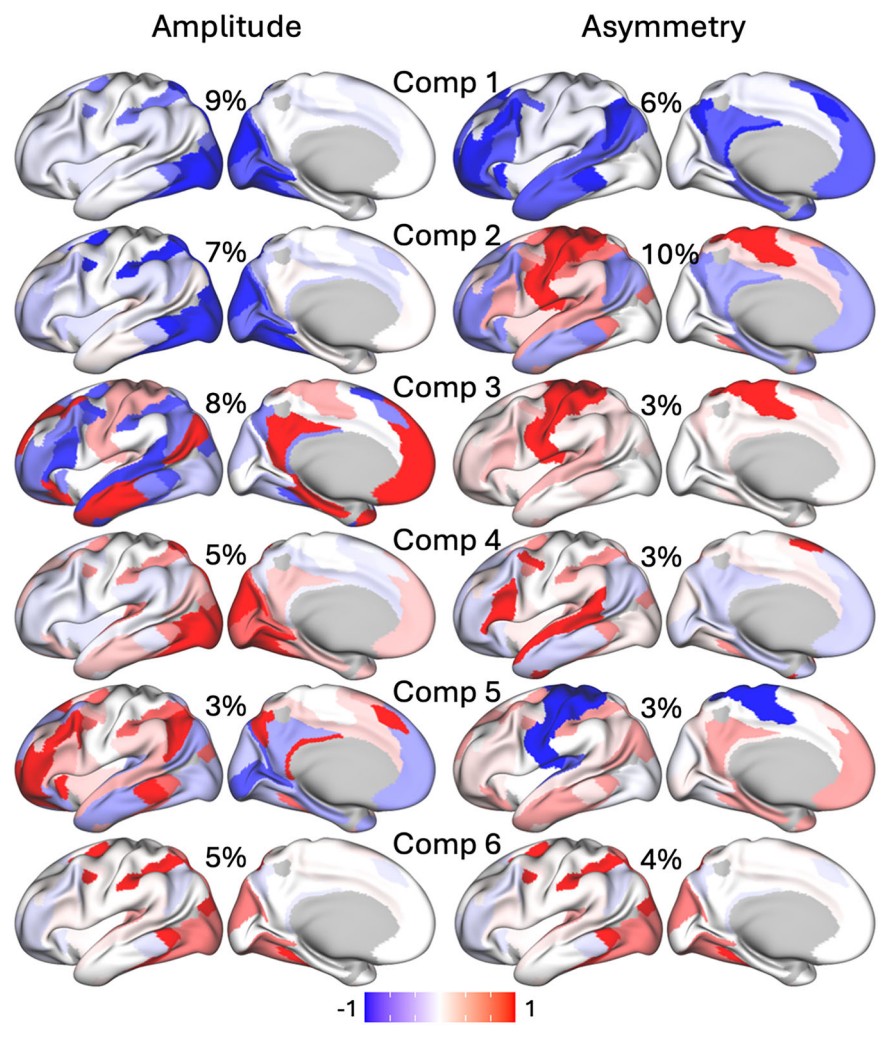

patterns. Our results highlight the role of asymmetry in specialized processing, particularly in complex cognitive tasks, where distinct hemispheric contributions may help manage cognitive load and optimize performance accuracy. The observed robust lateralization patterns in networks supporting language, motor function, and social cognition align with prior large-scale analyses of functional lateralization that identified four principal axes of asymmetry (symbolic communication, perception/action, emotion, and decision-making) using meta-analytic mapping[15].

Both greater fMRI signal amplitude and stronger hemispheric asymmetry were positively associated with task performance accuracy across multiple task epochs. These associations were especially robust in networks implicated in higher-order cognition and were most pronounced during relational processing, working memory, and story comprehension tasks. During the story comprehension task, asymmetry in the LAN network explained slightly more variance in performance than amplitude, suggesting a critical role for lateralized processing in language comprehension. The present findings suggest that both increased engagement (reflected in amplitude) and sustained hemispheric differentiation (reflected in asymmetry) contribute to better task outcomes, highlighting the functional relevance of maintaining both activation strength and lateralized organization across cognitive systems. However, our multivariate analysis indicates that the predictive contribution of asymmetry is secondary to that of amplitude. Specifically, a combined PLS model incorporating both measures did not outperform the amplitude-only model. This suggests that while asymmetry captures meaningful spatial characteristics of task-evoked brain activity, its association with behavior may, in large part, reflect underlying variation in activation strength.

Our findings support the left-hemisphere's dominant role in language processing, showing leftward asymmetry in language and frontoparietal networks during story tasks[36]. The right hemisphere dominance in attention and emotion networks during emotion tasks aligns with the role of the right hemisphere in social and emotional processing[37–39]. Motor tasks showed expected contralateral asymmetry in somatomotor and related networks[40,41]. Relational and working memory tasks revealed complementary hemispheric roles, with left-hemisphere rule-based and right-hemisphere spatial processing, reflected in DMN and FPN activations[6,42].

Consistent with our hypotheses, the amplitude robustly predicted task accuracy across working memory, language, and relational processing tasks, suggesting that greater bilateral neural engagement in task-relevant networks translates to better performance. This relationship underscores that greater fMRI signal amplitude is not simply an indicator of neural activity but a meaningful predictor of task accuracy, suggesting that bilateral and higher engagement within task-relevant networks may enhance the brain's functional capacity to support complex cognitive tasks. The emergence of overall neural engagement (fMRI amplitude) as a more consistent predictor of performance suggests that reductions in asymmetry, whether due to aging, neurodevelopmental variation, or psychiatric conditions, may be partially offset by stronger bilateral activation, consistent with the notion that dedifferentiation[26] can reflect adaptive reorganization rather than solely functional decline[22]. More broadly, our results imply that both amplitude and asymmetry should be considered when evaluating task-related dysfunction, as impaired lateralization may contribute to deficits in specific cognitive domains, whereas reduced engagement may have more global effects on performance. Future work in clinical populations will be

important to disentangle how these factors interact to shape cognitive outcomes.

On the other hand, asymmetry was a more constrained predictor of task accuracy, with significant effects mainly in specific epochs in working memory and relational processing tasks (only for 0- and 2-back, and match epochs). This observation suggests that hemispheric specialization provides targeted advantages in cognitive processes with limited capacity[43], such as working memory, that necessitate efficient resource allocation to optimize performance. Hemispheric asymmetry, in this context, may help offset these constraints by dividing processing loads between hemispheres.

PLS scores for component 2 of the amplitude-based accuracy predictions, which were associated with VIS and DAN network engagement, were significantly higher in male participants compared to females and demonstrated a marked decline with age across both the Discovery and Replication samples. In contrast, for asymmetry-based accuracy predictions, scores for components 1 and 3, which were predominantly associated with LAN and SMM networks, respectively, were significantly higher in female participants compared to males. These findings suggest that age and sex may modulate the neural architecture underlying task performance, with a more bilateral engagement in males for amplitude-driven accuracy and a more asymmetric involvement in females across LAN and SMM networks. This sex-based divergence in neural organization could reflect distinct processing strategies that contribute to task performance across task epochs.

The observed association between fMRI signal amplitude and hemispheric asymmetry is expected, as asymmetry metrics are inherently derived from differences in regional activation levels. Consequently, regions with greater overall activation may exhibit stronger asymmetry simply due to amplitude variability. This interdependence emphasizes the need to interpret asymmetry findings within the context of signal strength. In our data, the strongest amplitude–asymmetry associations emerged in lateral occipital, ventral temporal, inferior frontal, and inferior parietal regions, where greater neural engagement was consistently linked to reduced rightward asymmetry. These results suggest that amplitude may partly drive the magnitude and direction of observed lateralization patterns. Future work should employ analytic strategies that more effectively disentangle amplitude and asymmetry to clarify whether asymmetry contributes uniquely to cognitive performance beyond the effects of activation strength.

A potential limitation of our study is the restricted generalizability of the findings, given the narrow age range and inclusion of a predominantly healthy, young adult, predominantly white sample drawn from the HCP; this should be explicitly acknowledged, as lateralization patterns and their associations with cognitive performance may vary substantially across different age groups or clinical populations. Furthermore, our study uses a cross-sectional approach, which limits the ability to infer causal relationships between amplitude, asymmetry, and cognitive performance. Several tasks exhibited ceiling effects (e.g., motor, language) or near-chance performance (e.g., gambling, emotion), limiting their sensitivity to individual differences. As a result, brain-behavior associations may be attenuated or difficult to detect, and interpretations involving these tasks should therefore be made with caution. Differently, the relational and working memory tasks showed a broad distribution of accuracy scores, making them particularly well-suited for examining such brain-behavior associations. A key limitation of this study is the restricted scope of the HCP task battery, which does not include several well-established lateralized functions such as spatial attention, face perception, or emotional prosody, thereby limiting our ability to fully characterize the breadth of functional asymmetry across all task epochs.

In conclusion, our findings highlight that the mean fMRI signal amplitude was a stronger and more consistent predictor of task performance than hemispheric asymmetry. This suggests that the degree of neural engagement plays a more direct role in supporting cognitive performance. While asymmetry may capture important aspects of functional specialization, our results indicate that overall activation strength is a more robust marker of task-relevant neural processes. These findings underscore the importance of considering signal amplitude in future studies aiming to link brain activity with behavior.

## Methods

### Subjects

The data used in this study were sourced from the publicly accessible WU-Minn HCP 1200 Subjects data release (http://www.humanconnectome.org/). The scanning protocol was approved by the Human Research Protection Office (HRPO) at Washington University in St. Louis, where all participants included provided written informed consent, and all ethical regulations relevant to human research participants were followed. In all, 161 participants were excluded from the study due to incomplete image datasets. The remaining 989 participants (age: 28.8 ± 3.7 years; 607 females) were included in this study. No experimental procedures involving human subjects were conducted at the author's institutions.

### Task-fMRI paradigms

The HCP fMRI tasks paradigms are thoroughly detailed elsewhere[44]. They encompass a diverse set of tasks, including motor, language, social cognition, relational processing, working memory, gambling, and emotion processing, designed to probe distinct neural networks and capture the functional specialization and integration of the human brain (https://www.humanconnectome.org/hcp-protocols-ya-3t-imaging).

1. In the emotion task[45], participants matched either the emotional expression (angry or fearful) of a face at the top of the screen to one of two faces at the bottom or matched shapes in a similar manner. Each run contained 3 face epochs and three shape epochs, with six trials per epoch. Epochs were preceded by a 3-second cue and lasted 21 seconds, with stimuli presented for 2 seconds and a 1-second ITI. Each run consisted of six blocks and had a total duration of approximately 2 minutes and 16 seconds.

2. In the gambling task[46], participants played a card-guessing game where they predicted if the number on a mystery card (1–9) was higher or lower than 5 to win or lose money. They made their guess by pressing a button, and feedback (a green up arrow with "$1" for wins, a red down arrow with "−$0.50" for losses, or a neutral arrow for neither) was provided for 1 second. Each run contained 2 "mostly reward" and 2 "mostly loss" epochs (28 seconds each), with 8 trials per epoch, and 4 interleaved fixation epochs (15 seconds each). Each run of the task has a duration of approximately 3 minutes and 12 seconds.

3. The language task[32] includes 4 epochs of a math task and 4 epochs of a story task in each of two runs. Story epochs presented short auditory stories (5–9 sentences), followed by a forced-choice question about the story. The math task, designed to match the length of the story epochs (~30 seconds), involved solving auditory arithmetic problems. The math task was adaptive to ensure a consistent difficulty level across participants. Each run of the task has a duration of approximately 3 minutes and 57 seconds.

4. The motor task[40,47] involves visually cued movements like tapping fingers, squeezing toes, or moving the tongue. Each epoch lasted 12 seconds and was preceded by a 3-second cue. Each run included 2 epochs for each movement type (tongue, right hand, left hand, right foot, left foot) and 3 fixation epochs (15 seconds each). Each run of the task has a duration of ~3 minutes and 34 seconds.

5. In the relational processing task[48], participants viewed two pairs of objects, each with a distinct shape and texture. One pair was displayed at the top and the other at the bottom of the screen. During "relation" epochs, participants identified which feature (shape or texture) differed in the top pair, then determined if the bottom pair differed along the same dimension. During control "match" epochs, participants compared two objects at the top with one object at the bottom, using a word cue (either "shape" or "texture") to decide if the bottom object matched either of the top objects in that dimension. Stimuli were shown for 2800 ms, with a 400 ms inter-trial interval (ITI), and each epoch consisted of 5 trials. Each of the two task runs contained 3 "relation," 3 "match," and 3 "fixation" epochs, each lasting 16 seconds. Each run of the task has a duration of approximately 2 minutes and 56 seconds.

6. In the social cognition task, participants watched 23-second video clips of shapes (squares, circles, triangles) either interacting socially or moving randomly[49,50]. After each clip, participants judged whether the movements reflected social interaction, uncertainty, or no interaction. Each run included 5 video epochs (either social or random) and 5 fixation epochs (15 seconds each). Each run of the task has a duration of approximately 3 minutes and 27 seconds.

7. The working memory task (1) presents 4 categories of images (faces, places, tools, and body parts) across 8 epochs in 2 runs. Each category had a 0-back epoch (press a button for a target stimulus) and a 2-back epoch (press a button when the current stimulus matches one presented two steps earlier). Each epoch lasted 27.5 seconds and included ten 2.5-second trials, with 2 targets and 2–3 non-target lures (stimuli repeated in incorrect n-back positions). Each stimulus was displayed for 2 seconds, followed by a 500 ms ITI. Each run of the task has a duration of approximately 5 minutes.

## fMRI data

The fMRI data acquisition and image preprocessing methods of the HCP are described elsewhere[30,51]. Briefly, functional imaging was conducted using a 3.0 T Siemens Skyra scanner (Siemens Healthcare, Erlangen, Germany) equipped with a 32-channel coil. The imaging protocol employed a gradient echo-planar imaging (EPI) pulse sequence with a multiband factor of 8, a repetition time (TR) of 720 ms, an echo time (TE) of 33.1 ms, a flip angle of 52°, a matrix size of 104 × 90, and 72 slices, producing isotropic voxels of 2 mm[52,53]. For each subject and for each task, two full fMRI runs were acquired, one with left-right (LR) and one with right-left (RL) phase encoding directions. This within-task acquisition approach ensured consistent coverage across subjects and enabled more accurate correction of susceptibility-induced distortions using both runs rather than relying on a limited number of reversed-phase volumes. Preprocessing included gradient distortion correction, rigid-body realignment, field-map processing, and spatial normalization to the Montreal Neurological Institute (MNI) stereotactic space[30]. For the analysis, we utilized individual cross-run first-level contrasts of parameter estimates in CIFTI format from the WU-Minn HCP 1200 Subjects Data Release. We used 2 contrasts from the emotion task (faces and shapes, totaling 1978 images), 2 from the gambling task (reward and punish, 1978 images), 2 from the language task (story and math, 1978 images), 5 from the motor task (left hand, right hand, left foot, right foot, and tongue, totaling 4985 images), 2 from the relational processing task (rel and match, 1978 images), 2 from the social cognition task (social and random, 1978 images), and 2 from the working memory task (0-back and 2-back, 1978 images).

## fMRI asymmetry index

A normalized index of asymmetry was calculated by subtracting BOLD signal estimates in the right (R) hemisphere from those in the left (L) hemisphere for each task, contrast, and individual, following the formula:

$$\Delta = \frac{L - R}{|L| + |R|} \tag{3}$$

The denominator, $|L| + |R|$, represents the bilateral BOLD signal amplitude and reflects the combined activation from both hemispheres. This approach allowed for the quantification of asymmetry while accounting for the magnitudes of the estimates in both hemispheres[28]. A positive value of the asymmetry index indicates greater activity in the left hemisphere compared to the right, suggesting a leftward lateralization of the measured function. Conversely, a negative value suggests greater activity in the right hemisphere, indicating rightward lateralization. To confirm anatomical correspondence between hemispheres in the symmetric template space, we verified that each of the 32,492 left hemisphere vertices had a one-to-one mirrored counterpart in the right hemisphere by computing the correlation of their Cartesian coordinates (relative to the bounding box center), yielding values exceeding 0.995[28]. This threshold-independent

index on the cortical sheet provides a more nuanced view of interhemispheric lateralization, capturing subtle asymmetries that may be lost with threshold-dependent indices in the volumetric Cartesian space.

## PLS analysis

PLS regression projects both the predictor **X** and response **Y** matrices into lower-dimensional spaces and creates latent variables that represent linear combinations of the original variables while retaining the most meaningful variability in the data. By focusing on latent variables, PLS bypasses issues with highly correlated predictors and can handle situations where the number of predictors is larger than the number of observations. Furthermore, the latent variables can be interpreted as meaningful patterns in the data.

PLS regression was conducted in R to examine the relationships between fMRI signal amplitude, asymmetry, and task accuracy. Individual contrasts corresponding to 17 task epochs (faces, shapes, reward, punish, story, math, left hand, right hand, left foot, right foot, tongue, rel, match, social, random, 0-back, and 2-back) were averaged within 9 major networks (VIS1, VIS2, DMN, DAN, FPN, LAN, CON, AUD, SMM) to construct two matrices with 153 regressors, one for fMRI signal amplitude (**B**) and the other for asymmetry (**A**). To investigate whether the mean interhemispheric fMRI signal amplitude predicts asymmetry, we set **X** = **B** and **Y** = **A**.

The orbito-affective (ORA), ventral multimodal (VMM), and posterior multimodal (PMM) networks were excluded from the PLS prediction model due to their relatively small size and the diffuse, less task-specific nature of their activation patterns. As a result, including these networks could introduce significant variability and reduce the model's predictive accuracy. By prioritizing reliable networks with more distinct and localized functional roles, we aimed to increase the model's sensitivity to task-specific effects and improve the interpretability of the results.

To quantify how each functional network expressed the task-related amplitude/asymmetry patterns identified by PLS, we computed a set of $n = 12$ principal component (PC) scores by projecting network-level amplitude/asymmetry values onto each latent component. Specifically, for each of the $i = 1, \ldots, 9$ networks and each of the $n$ principal components, we calculated a weighted sum of amplitude/asymmetry values across epochs, using the weights defined by the loadings derived from the PLS model. Let $A_{i,j}$ denote the mean amplitude/asymmetry (averaged across subjects) for network $i$ and task epoch $j = 1, \ldots, 17$, and let $\omega_{i,j,n}$ be the loading of network $i$ and epoch $j$ on latent component $n$. The component score for network $i$ and component $n$ was computed as:

$$PC_{i,n} = \sum_{j=1}^{17} A_{i,j}\omega_{i,j,n} \tag{4}$$

PLS regression was also used to examine the relationships of amplitude and asymmetry with task accuracy. Specifically, individual accuracy scores corresponding to faces, shapes, reward, punishment, story, math, rel, match, social, random, 0-back, and 2-back epochs were arranged in another matrix with 12 variables (**C**). For predictions of accuracy based on amplitude or asymmetry, we set **Y** = **C** and **X** = **B** or **A**.

## Statistics and reproducibility

Participants were divided into discovery ($n = 504$) and replication ($n = 485$) subsamples matched by sex, age, and BMI using the sample.split function of the caTools R library. Vertex-wise statistical analyses of fMRI signal asymmetry were conducted in MATLAB 2022a. To account for potential confounds, we first regressed out age, using linear regression, followed by grand mean scaling to normalize the data across categorical variables (sex and race). The statistical significance of asymmetry was then assessed using vertex-wise t-tests, with a threshold $p < 0.05$, false discovery rate (FDR) corrected for multiple comparisons across 32,492 vertices on the cortical sheet. Statistical analyses within specific regions of interest (ROIs) were conducted using a $t$ test in R. To compare the fit of linear and nonlinear

models, we computed the Akaike Information Criterion (AIC) for each model, with lower AIC values indicating a better fit to the data. This approach allowed us to assess whether the addition of nonlinear terms, such as higher-order polynomials, provided a significant improvement over the simpler linear models.

## Reporting summary

Further information on research design is available in the Nature Portfolio Reporting Summary linked to this article.

## Data availability

HCP data are publicly available through the ConnectomeDB data management platform (https://db.humanconnectome.org/). Group-level statistical maps of functional asymmetry and bilateral amplitude in CIFTI space are publicly available on Figshare.com (https://doi.org/10.6084/m9.figshare.28204100).

## Code availability

The MATLAB code used to compute asymmetry in this study is available on Figshare.com (https://doi.org/10.6084/m9.figshare.29967877).

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

## Acknowledgements

We are thankful to Adam Thomas, PhD, Dustin Moraczewski, PhD, and Eric Earl, BS (National Institute of Mental Health Data Science and Sharing Team) for providing access to the HCP data on our servers. This study utilized the computational resources of the NIH HPC Biowulf cluster. (http://hpc.nih.gov). This work was done with support from the National Institute on Alcohol Abuse and Alcoholism (Y1AA-3009; ZIAAA000550).

## Author contributions

D.T. and N.D.V. designed the study and interpreted the data; D.T. developed code and statistical analysis; D.T. and N.D.V. wrote the manuscripts.

## Competing interests

The authors declare no competing interests.
