## [Transparent Peer Review file · Communications Biology]

Associations between fMRI Signal Amplitude, Hemispheric Asymmetry, and Task Performance

Corresponding Author: Dr Dardo Tomasi

Version 1:

Reviewer comments:

Reviewer #1

(Remarks to the Author)
Overall assessment

The main issue I found with the manuscript lies in the structure and clarity of the Introduction. In places, it reads more like a literature review, with the research questions remaining vague and not clearly articulated. As a result, it is difficult to grasp the central question the study aims to address. While there is some discussion on the relevance of studying asymmetry, the connection between this motivation and the specific results presented, including relationship with amplitude, is not clearly established.

Introduction

Overall, the authors need to provide a theoretical motivation why it is important to look at the amplitudes in relation to asymmetry, perhaps, taking a closer look at potential statistical relation between them. My understanding is that the main motivation here is the two competing hypotheses on functional lateralisation, segregation vs dedifferentiation, but I fail to see a direct link to the analyses of amplitudes, or how the authors are going to test the two hypotheses. I think this needs to be explained in a much clear form.

In more details:

I think the first two opening paragraphs, introducing the concepts of functional asymmetry, task, & segregation for efficiency are circular/repetitive if not convoluted, could be easily condensed by a factor of 2 without the loss of the content.

Third paragraph and up to "irrelevant or task-inappropriate activation in the contralateral hemisphere, leading to less efficient processing and poorer performance." – Why is all this discussion relevant for the study? What is the question here?

"These competing frameworks emphasize the need to examine how asymmetry relates to both task-evoked brain activity and behavioral performance." how are you going to test which is which?

"healthy adolescents/young adults with left-hemisphere language dominance (n=39) often display this pattern of asymmetry" – I *think* the authors by "this pattern" meant "right-hemisphere dominance for the spatial attention/facial recognition tasks", but this currently reads as "left-hemisphere language dominance is displayed as left-hemisphere language dominance"

"Yet, the generalizability and reproducibility of these findings remain unclear." – please clarify why, in order not to sound shallow.

"Surface-based analyses offer a precise approach" – I don't think this is such a great conceptual advance that is worth mentioning in the Intro. In any case, the second sentence in this paragraph is redundant, almost like an inverted form of the first one.

"a positive linear association between fMRI signal amplitude and hemispheric asymmetry," where does it come from?

“In parallel, we hypothesized that distinct task epochs..” – what is the novelty here, wasn’t this already demonstrated

Results

“were calculated and mapped to the 32,492 vertices” – needs explaining why both are important.

“fMRI signal asymmetry was task-specific” – how is “task-specificity” defined. I can see a lot of visual similarity between WM and RELATIONAL

“somatomotor (SMM) cortex” – this, and other partitioning into networks, comes out of blue. How you define networks needs to be mentioned somewhere.

“While motor epochs.. In contrast, functional..” these two sentences seem to belong to other section (Discussion,Intro?)

“ reflects the magnitude of asymmetry” - please add “average”, to be clear on what it means.

“We included all available HCP tasks..” – I wonder whether this and the following sentences could be moved to the very beginning of the section? I understand your point here, ie., that motor tasks or those with a motor component in response will have specific lateralisation due to spinal cord anatomy. But the way, it is presented here, is somewhat convoluted. Perhaps, you need to start with clearly stating the issue and then proceed with reporting the results.

“We used established functional network partitions.. “ – see my comment above, the network partitioning needs an earlier introduction

“In the language task, compared to math epochs...” – I am confused; there was already mentioning of this contrast in the previous paragraph. Are these different analyses? If this section presents a network-focused analyses (ie., “network X exhibits asymmetry in x,y,z tasks”), then this sentence is out of tune with this.

“We used the story > math contrast..” this perhaps belongs to Discussion.

“to reduce task-specific noise” – what is this?

“to decreased rightward asymmetry” – or increased leftward?

“amplitude and asymmetry appeared more modestly coupled” – or there was no asymmetry? In other words, the coupling between asymmetry and amplitude exists when there is an asymmetry? (in which case, this is a somewhat a trivial result).

“To also examine the relationship between asymmetry and the amplitude at the network level.. ” – If I understood this correctly, the paragraph above reports results where the vertex-wise correlations between amplitude and asymmetry computed first and then averaged per network (it’s a bit unclear to me how p-values were computed under this procedure); the current paragraph seems to report the result when the mean amplitude and asymmetry per network are computed first, and then their association is computed. Both report some sort of averaging, “network-level”, hence it is not clearly explained what additional information is gained.

Discussion

What do the results of this study have to say about issues raised in the introduction, such as dedifferentiation hypothesis, neurodevelopmental and psychiatric disorders?

Reviewer #2

(Remarks to the Author)

The authors have addressed my earlier points.

Reviewer 1

Comment 1: “Unlike traditional volumetric methods, “ – this is a very strange argument in favour of surface-based analyses. One can accept that inter-subject alignment to the surface template space can be more accurate due to 2d nature of the optimisation problem, but advantage in terms of possession of a symmetric template – this is not correct; creation of a symmetric volumetric template is a trivial exercise (or one can use symmetric MNI templates, they do exist).

Response: We agree with this, and we have removed the sentence suggesting that the availability of a symmetric template is a unique advantage of surface-based methods. Our emphasis remains on the improved inter-subject alignment afforded by surface-based registration.

Comment 2: “Scans were collected using both left-right (LR) and right-left (RL) phase encoding directions to enhance data reliability.” – unclear context. Did orientations vary across subjects, across-tasks, within-task? Or is it just a few volumes with the opposite direction to feed into topup?

Response: Thank you for pointing this out. We have revised the text for clarity. It now reads: “For each subject and for each task, two full fMRI runs were acquired, one with left-right (LR) and one with right-left (RL) phase encoding directions. This within-task acquisition approach ensured consistent coverage across subjects and enabled more accurate correction of susceptibility-induced distortions using both runs rather than relying on a limited number of reversed-phase volumes.” (lines 564-568).

Comment 3: “The correspondence of 32,492 vertices in the left hemisphere with those in the right hemisphere” – this and the next sentence. I am not sure I understood the logic of the correlational analysis, perhaps, the co-authors could clarify this a bit; doesn't left-right symmetric space presume one-to-one correspondences (homologues) between left and right vertices?

Response: We agree that the symmetric template space presumes one-to-one correspondences between left and right hemisphere vertices. The sentence in question was intended to report a geometric validation of that symmetry, by confirming that vertex coordinates were mirrored as expected across hemispheres (with correlation coefficients > 0.995). To clarify, we have revised the text to make clear that this initial analysis served only to verify anatomical correspondence in the template space, not to establish it.

“To confirm anatomical correspondence between hemispheres in the symmetric template space, we verified that each of the 32,492 left hemisphere vertices had a one-to-one mirrored counterpart in the right hemisphere by computing the correlation of their

Cartesian coordinates (relative to the bounding box center), yielding values exceeding 0.995" (lines 588-592).

Comment 4: "The ORA, VMM, and PMM networks."— due to their professional specialisation, some readers may want to read Methods before Results. Perhaps, spelling out the names of these networks could help them here.

Response: Thank you for the suggestion. We agree that spelling out the network names improves clarity, especially for readers consulting the Results before the Methods. We have revised the text to include the full names on first mention: the Orbital-Affective (ORA), Ventral Multimodal (VMM), and Posterior Multimodal (PMM) networks. (lines 609 and 610)

Comment 5: "For amplitude-based predictions of asymmetry we set $X = B$ and $Y = A$ " – was this motivated somehow? Why do we want to look at this, any relation to a specific hypothesis?

Response: We agree that additional context was needed to clarify the rationale. This analysis was motivated by the hypothesis that higher bilateral activation may drive greater hemispheric differentiation, as reflected in asymmetry patterns. We have revised the sentence to read:

"To investigate whether the mean interhemispheric fMRI signal amplitude predicts asymmetry, we set $X = B$ and $Y = A$." (lines 607 and 608)

Comment 6: The motivation for analyses using network partitioning wasn't made clear. Why is this more useful than just calling their anatomical locations?

Response: This approach allowed us to interpret asymmetry patterns in terms of coherent functional systems rather than isolated anatomical regions, facilitating comparisons across functional domains. In the Results, we now provide the following justification:

"We used established functional network partitions to organize regional fMRI signals, enabling a structured analysis of hemispheric asymmetry across the brain." (lines 198 and 199)

Comment 7: Figure 2 – do I understand this correctly that for all tasks, except for left foot and hand and (possibly) language, left asymmetry in the sensorimotor cortex is the effect of a response limb?

Response: Thank you for this insightful comment. Yes, your interpretation is largely correct. For most tasks, leftward asymmetry in the sensorimotor cortex likely reflects the use of the right hand (or foot) for responses, consistent with the contralateral

organization of motor control. The exceptions are the left foot and left-hand motor tasks, which naturally produce right-lateralized activation, and the language task, which shows leftward asymmetry due to well-established hemispheric specialization. We have clarified this interpretation in Results:

“For all tasks, except those involving left limb movements and possibly language, leftward asymmetry in the sensorimotor cortex likely reflects activation related to the contralateral (right) response limb (Figure 2)” (lines 157-159)

Comment 8: “...with rightward asymmetry in the visual and SMM cortex” –I can only see this for visual cortex in Figure 2 but not for SMM. Does this come from subtracting between tasks conditions, which is not shown in Figure 2?

Response: Thank you for your observation. We have clarified that the rightward asymmetry in the somatomotor (SMM) cortex arises from contrasts between task and control epochs, which are presented in Supplementary Figure S1. Figure 2 focuses on overall task epochs, so these more specific contrasts are shown separately for clarity.

Comment 9: “A similar asymmetry pattern appeared when contrasting relational” – similar to the above, not clear where I should look at to appreciate this fact.

Response: See response to Comment 8.

Comment 10: Figure 3 – what do percentages (10.87 and 8.33) signify?

Response: The percentages (10.87 and 8.33) in Figure 3 represented the relative root-mean-square (RMS) difference between the matrices. To avoid confusion, we have removed these labels from the figure.

Comment 11: “In the language task, story epochs—compared to math epochs” is not it concerning to the authors that asymmetry is defined with respect to a competing task rather than to “rest”?

Response: Thank you for raising this point. We have added the following justification to clarify our approach:

“We used the story > math contrast to isolate language-specific processes while controlling for auditory, attentional, and working memory demands; this contrast has been validated in prior work as a reliable marker of language-related lateralization⁴⁶.” (lines 216-219)

Comment 12: “To identify spatial patterns linking signal amplitude” – I don’t remember how signal amplitude was measured.

Response: As defined in Equation [2], bilateral amplitude reflects the average fMRI signal across corresponding vertices in both hemispheres. This measure was used to

summarize overall activation strength independent of lateralization. We have revised the sentence for clarity as:

“To identify spatial patterns linking mean bilateral amplitude (Eq [2])...” (lines 239-243)

Comment 13: “These findings indicate that the amplitude-asymmetry relationship varies systematically across cortical territories” – ... within the correlation range of .5 - .7, according to Figure 4a. How significant is this?

Response: The correlations range from approximately 0.5 to 0.7 reflect highly robust associations given the large sample size ($N = 989$). Specifically, a correlation coefficient greater than 0.5 corresponds to $P < 2.2 \times 10^{-16}$. We have now added this information to the Figure 4 legend to clarify the statistical significance of the findings. (line 262)

Comment 14: “this ranking was used to form 100 groups of 10 individuals with similar amplitude levels” – not sure why you would need this partitioning. Why couldn't the associations be studied without grouping?

Response: We used this partitioning approach to visualize how the relationship between amplitude and asymmetry varied across the full amplitude distribution while reducing inter-individual variability. Grouping participants into bins of similar amplitude levels enabled us to average asymmetry measures within each group, thereby improving the signal-to-noise ratio and facilitating the identification of systematic trends across the amplitude spectrum.

Comment 15: “The correlation analysis revealed that the amplitude-asymmetry association” – I wasn't sure what distinguishes this analysis from the earlier analysis of the same association.

Response: the sentence was removed.

Comment 16: “For epochs within the social and gambling tasks, the amplitude regressors successfully predicted asymmetry primarily in the LAN” – I wasn't sure how the authors mapped between latent variables that were found in PLS and concrete tasks/networks.

Response: We thank the reviewer for pointing this out. To clarify, we mapped the latent task components obtained from the PLS analysis onto functional networks by computing a set of component scores that summarize how each latent component weighted task-related asymmetry patterns across networks. Specifically, for each network, we took the mean asymmetry values across subjects and projected them onto each latent component using the corresponding loadings from the PLS model. This produced a matrix of component scores, where each score represents the extent to which a

particular network expressed the spatial asymmetry pattern associated with a given latent task component. This allowed us to identify which networks (e.g., LAN) most strongly expressed task-related asymmetry patterns captured by the PLS model. We have now clarified this procedure in the revised Methods section (Eq [3]). (lines 615-624)

Comment 17: “As shown in Figure 5b–c, greater signal amplitude and asymmetry were significantly correlated” - there is no differentiation between amplitude and asymmetry in plots b-c; Figure 5 says “Task accuracy and asymmetry”.

Response: We appreciate the reviewer’s careful observation. We agree that in the original version of Figure 5b–c, the plots did not distinguish between signal amplitude and asymmetry; instead, only asymmetry was shown. In response to this, we have revised the figure. The new version of Figure 5 (panels b-e) now includes Pearson correlation matrices showing associations of accuracy with both amplitude and asymmetry. The figure legend has been updated accordingly to read: “Task accuracy vs signal amplitude and asymmetry.” (lines 314-322)

Comment 18: “These results suggest that both the strength and lateralization of task-evoked neural responses are meaningful contributors” – this pictures amplitude and asymmetry as two relatively independent factors, which they are not; hence, it is not surprising at all that, if one accounts for accuracy, the second will also be. I guess this just says that the shared variance between lateralisation and amplitude bear information on accuracy. In short, some estimation of how much shared, how much unshared is needed.

Response: We agree that the original sentence may have implied an undue independence between amplitude and asymmetry, despite their inherent correlation. To avoid this misinterpretation, we have removed the sentence from the manuscript. We now focus on describing the associations more cautiously, without implying independence. (lines 38-39 and 461-471)

Reviewer 2

Comment 1: The theoretical motivation for the study is not clearly articulated, and the data appear ill-suited to address the hypotheses posed. As a result, the contribution to the existing literature is limited.

Response: We appreciate the reviewer’s comment and have revised the manuscript to clarify the theoretical motivation and better articulate the relevance of our data to the hypotheses posed. The study is grounded in longstanding theories of hemispheric specialization and neural efficiency, which suggest that functional asymmetry enables selective recruitment of specialized cortical regions, promoting efficient cognitive processing. While prior work has linked lateralization to performance in individual tasks

(e.g., language, motor), few studies have systematically compared asymmetry across multiple domains using uniform methodology. (lines 60-67, 83-86, and 92-93) By leveraging the high-quality, surface-based task-fMRI data from the Human Connectome Project (HCP), our study directly addresses this gap. The HCP dataset is uniquely suited for this purpose, providing a broad battery of lateralized tasks, consistent preprocessing, and performance metrics that allow us to examine associations between task-related asymmetry and behavior. (lines 104-111) While we did not find strong or consistent associations between asymmetry and performance accuracy, our results reveal stable and domain-specific patterns of functional asymmetry, supporting the view that lateralization is a reproducible feature of task-evoked brain activity.

Comment 2: The inclusion of all available tasks and contrasts in the asymmetry analysis raises concerns. For instance, motor tasks (e.g., hand, foot, tongue movements) primarily engage primary motor cortices governed by white matter decussations—a fundamentally different organizational principle than the hemispheric specialization observed in associative cortices. This distinction is not acknowledged or theoretically justified.

Response: We agree that primary motor functions, such as hand, foot, and tongue movements, are mediated by well-established contralateral control via white matter decussations, and therefore reflect a different organizational principle than the hemispheric specialization typically observed in associative cortices. Our rationale for including these tasks was twofold: first, to examine the consistency of asymmetry patterns across all major functional domains represented in the HCP task battery; and second, to provide a reference point for interpreting the magnitude and reliability of asymmetry in associative tasks relative to those with well-characterized lateralization (e.g., motor and language). We acknowledge, however, that this distinction between structural decussation and functional specialization was not clearly articulated in the original manuscript. We now address this in the revised Vertex-wise interhemispheric asymmetry subsection of Results:

“For all tasks, except those involving left limb movements and possibly language, leftward asymmetry in the sensorimotor cortex likely reflects activation related to the contralateral (right) response limb (Figure 2). While motor epochs (e.g., hand, foot, tongue movements) elicit lateralized activation due to anatomical decussations, where descending motor pathways cross the midline at the medullary pyramids, this form of lateralization is driven by structural wiring rather than hemispheric specialization per se. In contrast, functional asymmetry in associative cortices reflects differential engagement of the hemispheres based on task demands and cognitive processing strategies. We included all available HCP tasks and contrasts in our analysis to provide a comprehensive characterization of functional asymmetry across both sensorimotor and higher-order functional domains. Including well-established motor asymmetries served

as a benchmark for assessing the strength and reproducibility of lateralization in more complex tasks, enabling systematic comparisons across brain systems and task types within the same dataset.” lines(157-174)

Comment 3: Moreover, the classification of tasks such as “relational” and “gambling” as cognitive domains is questionable, as these are simply tasks included in the HCP protocol. In contrast, well-established lateralized functions such as spatial attention, face perception, and emotional prosody are not examined due to their absence in the dataset. This limits the study’s ability to meaningfully address mechanisms of lateralization.

Response: Our categorization of “relational” and “gambling” tasks as cognitive domains was not intended to imply that these tasks define entire cognitive domains in a formal theoretical sense. Rather, we adopted the task labels provided by the HCP as shorthand for the primary cognitive processes each task was designed to engage (e.g., relational reasoning and reward processing, respectively). We fully acknowledge that the HCP task battery does not encompass all classic lateralized functions, such as spatial attention, face perception, or emotional prosody, and this limitation is now more explicitly discussed in the revised Discussion. (lines 483-487) Nonetheless, our aim was to leverage the standardized and high-quality HCP dataset to evaluate reproducible patterns of functional asymmetry across a broad, albeit constrained, range of tasks. The revised manuscript refrains from equating tasks as cognitive domains.

Comment 4: Some task contrasts appear poorly chosen. For example, using a math task as a control for story listening to assess language processing is problematic and should be reconsidered or better justified.

Response: We thank the reviewer for highlighting the potential limitations of specific task contrasts, particularly the use of math as a control condition for story listening in the language task. This contrast was selected based on the design of the HCP task battery, which pairs story comprehension with an active control (math problems) to control for auditory input and working memory demands. While we acknowledge that using a resting or non-linguistic baseline might more directly isolate language-specific processes, the math condition serves to subtract general task-related activation, highlighting regions more selectively involved in narrative comprehension. This design has been validated and widely used in prior HCP-based language studies, and we retained it to ensure comparability and consistency with the broader literature. We have added clarification to the manuscript to better justify this choice in Networks and tasks ranked by asymmetry. (lines 216-219)

Comment 5: Task accuracy is a central aspect of the study and potentially its most valuable contribution. However, Figure 5a reveals that many tasks suffer from ceiling effects or near-random performance, undermining their utility for correlational analyses.

Only the “Relations” task shows a performance distribution suitable for such analysis, as reflected in Figure 5b. This is a significant limitation.

Response: We appreciate the reviewer’s careful attention to the task performance data and agree that ceiling effects (e.g., in Motor and Language tasks) and near-chance performance (e.g., in Gambling) limit the interpretability of brain–behavior correlations in several conditions. As noted by the reviewer, the Relational task (also the working memory task) offered appropriate distributions for examining associations with asymmetry, which we highlighted in Figure 5b. (lines 305-309) We have explicitly acknowledged this limitation in the revised Results and Discussion sections and clarified that performance-related analyses should be interpreted with caution for tasks exhibiting restricted behavioral variance. (lines 478-481) Despite this constraint, we believe our findings in the Relational and Working Memory tasks demonstrate the utility of integrating behavioral data with asymmetry measures and underscore the need for task designs that allow more dynamic performance ranges in future studies of lateralization.

Comment 6: The observed relationship between fMRI signal amplitude and asymmetry may be expected, given that asymmetry is derived from amplitude differences. The possibility that greater asymmetries occur in regions with higher activation should be acknowledged and discussed.

Response: Indeed, as asymmetry is calculated from the difference in fMRI signal amplitude between hemispheres, it follows that regions with higher activation are more likely to exhibit greater asymmetry. We now explicitly acknowledge this dependency in the first two sentences to the introduction paragraph in Results and have clarified that the observed relationship between signal amplitude and asymmetry may, in part, reflect this coupling (lines 235-243). At the same time, we note that not all regions with high activation necessarily show strong lateralization, and vice versa, suggesting that asymmetry is not solely driven by overall amplitude but also by the distribution of activity across hemispheres. Nonetheless, we agree that this relationship warrants careful interpretation, and we now include a discussion of this potential confound and its implications for our findings. (lines 461-471)

Comment 7: The rationale for the selection of network partitions is unclear. Additionally, the inclusion of the default mode network (DMN), which is theoretically unrelated to task-specific activation, warrants further explanation.

Response: In our study, we used a network cortical parcellation that offers a fine-grained and functionally grounded framework for summarizing activation patterns across the cortex. This atlas is widely used in recent neuroimaging studies and aligns well with the organization of large-scale functional systems observed in both rest and task states. Although the DMN is traditionally associated with internally directed processes and deactivates during many externally focused tasks, it can nonetheless

exhibit lateralized engagement depending on the cognitive context, particularly for tasks involving autobiographical memory, narrative comprehension, or mentalizing. Including the DMN in our analysis allowed us to examine whether such lateralized patterns emerge across different task domains and ensured comprehensive coverage of cortical networks. We have clarified this rationale in the revised Results section. (lines 198-208)

Comment 8: Introduction: In paragraph 2, clarify the distinction between cognitive functions (as operationalized by tasks and contrasts) and brain regions (as defined by activation patterns). In the final paragraph, provide references for asymmetry claims across cognitive domains and explain the rationale for hypothesizing a link between signal amplitude and task accuracy.

Response: We thank the reviewer for the helpful suggestions regarding clarity in the Introduction. In paragraph 2, we have revised the text to more clearly distinguish between cognitive functions, which are engaged through specific task paradigms and contrasts, and the brain regions that are identified based on differential activation patterns. This distinction is now explicitly stated to avoid conflating task design with neural localization. Additionally, we have added references to support domain-specific asymmetry patterns. (lines 60-67)

Comment 9: Results: The radar chart in Figure 2b is difficult to interpret. Please include a clearer explanation of how to read the chart, especially the meaning of the concentric circles.

Response: We thank the reviewer for noting the lack of clarity in interpreting Figure 2b. To improve interpretability, we have revised the corresponding text in the Results section to provide a clearer explanation of how to read the radar chart, particularly the meaning of the concentric circles. Specifically, we now state: "The radar plot in Figure 2b summarizes the asymmetry index values across functional networks for each task contrast. Each axis corresponds to a functional network, and the distance from the center reflects the magnitude of asymmetry, with concentric circles representing increasing values (e.g., -0.30, -0.15, 0.00, 0.15, 0.30) that indicate greater leftward lateralization." (lines 166-174)

Comment 10: Discussion: The initial discussion of asymmetry findings largely reiterates known results. The subsequent analysis of signal amplitude and task accuracy is more novel but is not reflected in the title. Given the limitations in task accuracy, the interpretation of these findings should be approached with caution. The limitations section could be expanded to include additional methodological considerations and suggestions for future research.

Response: We appreciate the reviewer's thoughtful feedback. In response, we have revised the Discussion to more clearly distinguish our findings from established results,

emphasizing the novelty of our approach, particularly the joint analysis of signal amplitude, asymmetry, and task performance across multiple tasks. (lines 425-430, and 461-471) While the observed associations involving task accuracy are limited by ceiling and floor effects in some tasks, we have explicitly acknowledged this constraint and now more cautiously interpret these results. We have also expanded the Limitations section to include additional methodological considerations, such as the reliance on a fixed set of tasks in the HCP dataset, potential confounds related to activation magnitude influencing asymmetry, and the need for replication in independent samples with broader task coverage (lines 478-487). The revised title of the manuscript, “Associations between fMRI Signal Amplitude, Hemispheric Asymmetry, and Task Performance”, better captures the contribution of fMRI signal amplitude to task accuracy.

Reviewer 3

Comment 1: Both greater fMRI signal amplitude and stronger hemispheric asymmetry were positively associated with task performance accuracy across multiple cognitive domains, with amplitude showing a stronger association with performance than asymmetry. However, as also shown by the authors, amplitude predicted asymmetry, i.e. these functional brain measures were not independent (and indeed were calculated from the same data). The question then arises whether asymmetry predicts performance over and above the effect of amplitude. A key finding, that I feel did not receive appropriate prominence in the paper, was therefore that ‘a combined PLS model incorporating both amplitude and asymmetry metrics did not outperform the amplitude-only model.’ From this the authors conclude that the contribution of asymmetry to performance is secondary to amplitude, but they seem unwilling to make a more conservative interpretation: that the apparent contribution of asymmetry to performance may be entirely driven by amplitude, and not due to asymmetry at all. This finding contradicts the main message of the paper as reflected in its current abstract and discussion. The relevance of asymmetry to task performance appears not clearly supported by the data.

Response: Thank you for this insightful comment. We agree that the relationship between amplitude and asymmetry deserves careful interpretation, particularly given their shared derivation from the same underlying fMRI data. In the revised manuscript, we have clarified this issue in both the Results and Discussion sections. (lines 235-243, and 461-471) While our findings show that both amplitude and asymmetry are positively associated with task performance, the combined PLS model did not improve prediction relative to amplitude alone. This suggests that the behavioral relevance of asymmetry may be largely accounted for by activation strength. We now more explicitly acknowledge this possibility in the Discussion, stating that the contribution of asymmetry to performance may be secondary to, or even dependent on, amplitude (lines 461-471).

We have also revised the Abstract to more conservatively reflect this interpretation and removed language that may have overstated the independent predictive value of asymmetry (lines 38-39). We appreciate the reviewer's attention to this key point, which has helped us refine the presentation and interpretation of our findings.

Comment 2: It was not always clear to me, for each specific analysis, whether the amplitude of fMRI signal was calculated directly from task data, or from task contrast data. For example, the Figure 4 legend refers to 'prediction of asymmetry based on mean bilateral BOLD amplitude across 17 task contrasts' but does this mean 17 task epochs? There were 11 task contrasts as I understood based on fig 3a. Can the authors please check through the paper and ensure that this aspect is always correct and explicit? When task contrasts were used, it is not clear to me that a difference in BOLD response between two tasks is necessarily a good measure of the efficiency of neural engagement for just one of those tasks, which seemed to be a central hypothesis from the introduction.

Response: Upon review, we confirm that the majority of analyses, particularly those shown in Figures 2, 3b, 4, 5, 6, and 7 are based on task epoch beta estimates, not task contrast values. Figure 3a, however, includes task contrast-based asymmetry rankings, as now clarified in the revised legend.(line 229) Figure 4 reports results based on 17 task epochs, and its legend has been updated accordingly. (line 261) We agree that contrasts between two task conditions can complicate the interpretation of neural engagement in a single condition. For this reason, we focused primarily on task epochs when analyzing the relationship between BOLD signal amplitude, asymmetry, and behavioral accuracy. Where contrasts were used, such as in supplemental figures or exploratory analyses, we now explicitly state this in the text and figure legends to avoid confusion.

Comment 3: In figure 3, the order of networks from top to bottom differs between parts a and b. Can this be made consistent?

Response: In Figure 3, the vertical order of networks differs between panels because both 2D matrices are independently ranked along the vertical axis by root-mean-square (RMS) asymmetry values. The horizontal axis is ordered by mean asymmetry across task contrasts (Figure 3a) or task epochs (Figure 3b). This ranking strategy highlights distinct patterns in asymmetry magnitude and direction.

Comment 4: The distributions of some performance measures appear highly non-normal (Figure 5a). Can the authors provide reassurance on their suitability for the statistical methods and models, or perhaps use permutations to assess significance of associations?

Response: While some performance measures do exhibit skew or non-normality (as shown in Figure 5a), we believe our PLS statistical analyses are robust to such deviations, as PLS does not assume normally distributed variables.

Reviewer #1

Overall comment: “The main issue I found with the manuscript lies in the structure and clarity of the Introduction. In places, it reads more like a literature review, with the research questions remaining vague and not clearly articulated. As a result, it is difficult to grasp the central question the study aims to address. While there is some discussion on the relevance of studying asymmetry, the connection between this motivation and the specific results presented, including relationship with amplitude, is not clearly established.”

Response: we revised the introduction to streamline the background, reduce the literature-review tone, and explicitly link the motivation for studying brain asymmetry to our specific hypotheses and the results we present, including the relationship with amplitude measures.

Introduction Overall comment: “the authors need to provide a theoretical motivation why it is important to look at the amplitudes in relation to asymmetry, perhaps, taking a closer look at potential statistical relation between them. My understanding is that the main motivation here is the two competing hypotheses on functional lateralisation, segregation vs dedifferentiation, but I fail to see a direct link to the analyses of amplitudes, or how the authors are going to test the two hypotheses. I think this needs to be explained in a much clear form.”

Response: We revised the manuscript to more explicitly connect our analyses to the two competing hypotheses on functional lateralization by explaining how amplitude asymmetries may reflect either more specialized (segregated) or less distinct (dedifferentiated) functional organization. Additionally, we clarified the statistical approach used to assess the relationship between amplitude and asymmetry, highlighting how these analyses provide a direct test of the two hypotheses.

Comment 1: “I think the first two opening paragraphs, introducing the concepts of functional asymmetry, task, & segregation for efficiency are circular/repetitive if not convoluted, could be easily condensed by a factor of 2 without the loss of the content.”

Response: We thank the reviewer for this suggestion. We have condensed the text, removed redundancy while preserving the key concepts regarding functional asymmetry, task-specific processing, and segregation for cognitive efficiency (lines 42-62).

Comment 2: “Third paragraph and up to “irrelevant or task-inappropriate activation in the contralateral hemisphere, leading to less efficient processing and poorer performance.” – Why is all this discussion relevant for the study? What is the question here?”.

Response: The paragraph was intended to provide a comprehensive background on the clinical and cognitive relevance of hemispheric asymmetry, situating our study within the broader literature on lateralization. We have removed the controversial paragraph to simplify the introduction.

Comment 3: “These competing frameworks emphasize the need to examine how asymmetry relates to both task-evoked brain activity and behavioral performance.’ how are you going to test which is which?”.

Response: We thank the reviewer for this important point. In response, we reworded the original paragraph to clarify the competing interpretations of reduced hemispheric asymmetry (lines 63-68). Our study was designed to empirically test these competing frameworks by linking individual differences in hemispheric asymmetry to both task-evoked neural activity and behavioral performance. Specifically, we examined whether reduced asymmetry was associated with preserved or enhanced task performance, which would support the compensatory recruitment hypothesis, or whether it correlates with poorer performance, consistent with the dedifferentiation framework. By directly relating lateralization metrics to behavioral outcomes in a controlled cognitive task, we aim to clarify which interpretation better explains variability in healthy adults.

Comment 4: “healthy adolescents/young adults with left-hemisphere language dominance (n=39) often display this pattern of asymmetry’ – I *think* the authors by “this pattern” meant “right-hemisphere dominance for the spatial attention/ facial recognition tasks”, but this currently reads as “left-hemisphere language dominance is displayed as left-hemisphere language dominance”.

Response: We thank the reviewer for pointing out this ambiguity. We have clarified the sentence to specify that “this pattern of asymmetry” refers to right-hemisphere dominance for spatial attention and facial recognition tasks, rather than left-hemisphere language dominance (lines 42-46).

Comment 5: “Yet, the generalizability and reproducibility of these findings remain unclear.’ – please clarify why, in order not to sound shallow.”

Response: The sentence was removed.

Comment 6: “ ‘Surface-based analyses offer a precise approach’ – I don’t think this is such a great conceptual advance that is worth mentioning in the Intro. In any case, the second sentence in this paragraph is redundant, almost like an inverted form of the first one.”

Response: To improve clarity and avoid redundancy, we removed the second sentence in this paragraph.

Comment 7: “ ‘a positive linear association between fMRI signal amplitude and hemispheric asymmetry,’ where does it come from?”

Response: We thank the reviewer for pointing this out. We now support this hypothesis by citing Jansen (2006), which provides empirical evidence for a positive linear association between fMRI signal amplitude and hemispheric asymmetry (lines 84-85).

Comment 8: “ ‘In parallel, we hypothesized that distinct task epochs..’ – what is the novelty here, wasn’t this already demonstrated”.

Response: We thank the reviewer for pointing this omission. We revised the sentence to clarify the novelty of our approach. The manuscript now emphasizes that, although canonical lateralization profiles have been described for various domains, it remains unclear how reproducibly these patterns emerge within individuals across separate datasets. To address this, we hypothesized that distinct task epochs would yield consistent asymmetry profiles and tested this in two matched Human Connectome Project subsamples, enabling a direct, cross-cohort reproducibility assessment (lines 87-94).

Comment 9: “were calculated and mapped to the 32,492 vertices” – needs explaining why both are important.”

Response: “were calculated” was removed

Comment 10: “‘fMRI signal asymmetry was task-specific’ – how is “task-specificity” defined. I can see a lot of visual similarity between WM and RELATIONAL”.

Response: To avoid ambiguity, we have revised the wording in the manuscript: rather than “task-specific,” we now state that “fMRI signal asymmetry varied across tasks” (line 132), which more accurately reflects the observed differences without implying strict exclusivity between tasks.

Comment 11: “ ‘somatomotor (SMM) cortex’ – this, and other partitioning into networks, comes out of blue. How you define networks needs to be mentioned somewhere.”

Response: We have now clarified in the Methods that the network labels, including somatomotor (SMM) cortex, follow the Cole-Anticevic Brain Network partition (lines 134-135). This reference provides the rationale for the network definitions used throughout the manuscript.

Comment 12: “ ‘While motor epochs.. In contrast, functional..’ these two sentences seem to belong to other section (Discussion,Intro?)”

Response: These sentences were removed from Results.

Comment 13: “ ‘ reflects the magnitude of asymmetry’ - please add ‘average’, to be clear on what it means”.

Response: ‘Average’ was added as requested.

Comment 14: “ ‘ We included all available HCP tasks..’ – I wonder whether this and the following sentences could be moved to the very beginning of the section? I understand your point here, ie., that motor tasks or those with a motor component in response will have specific lateralisation due to spinal cord anatomy. But the way, it is presented here, is somewhat convoluted. Perhaps, you need to start with clearly stating the issue and then proceed with reporting the results.”

Response: We thank the reviewer for the suggestion. We have revised the opening paragraph of the Vertex-wise interhemispheric asymmetry subsection for clarity (lines 127-131). It now reads:

"To comprehensively characterize functional asymmetry across sensorimotor and higher-order domains, we included all available HCP tasks and contrasts in our analysis. This approach allowed us to benchmark well-established motor asymmetries and systematically compare the strength and reproducibility of lateralization across brain systems and task types within the same dataset."

This rewording moves the rationale to the beginning of the section and clarifies the purpose before presenting the results.

Comment 15: “ ‘ We used established functional network partitions.. ‘ – see my comment above, the network partitioning needs an earlier introduction”

Response: see response to Comment 11.

Comment 16: “ ‘ In the language task, compared to math epochs...” – I am confused; there was already mentioning of this contrast in the previous paragraph. Are these different analyses? If this section presents a network-focused analyses (ie., “network X exhibits asymmetry in x,y,z tasks”), then this sentence is out of tune with this.”

Response: The previous paragraph describes voxel-wise asymmetry analyses, whereas the sentence in question reports results from a network-level analysis based on the Cole-Anticevic Brain Network Partition. While both analyses use the same language

versus math contrast, the former identifies localized asymmetries at the voxel level, and the latter summarizes how these effects are expressed across functional networks. To avoid confusing the reader we added a short explanatory statement indicating that this is a network-based analysis whereas the prior paragraph reported on a voxel wise analyses.

Comment 17: “ ‘We used the story > math contrast..’ this perhaps belongs to Discussion.”

Response: We decided to retain it in the Results section of the voxel-wise analysis (lines 143-145) to address a prior reviewer’s comment requesting that the contrast be explicitly described alongside the corresponding findings. Keeping it here ensures clarity for readers regarding the specific comparison underlying the observed regional asymmetries.

Comment 18: “ ‘to reduce task-specific noise’ – what is this?”

Response: By “task-specific noise,” we refer to variability in activation patterns that is unique to individual tasks or specific stimulus conditions, which may obscure general relationships between fMRI amplitude and hemispheric asymmetry. Averaging the vertex-wise correlations across all 17 task epochs allows us to minimize these task-specific effects and reveal consistent, task-general associations between amplitude and asymmetry. We have clarified this point in the revised manuscript (lines 217-219).

Comment 19: “ ‘to decreased rightward asymmetry’ – or increased leftward?”

Response: We thank the reviewer for pointing this out. The observed effect reflects a shift toward more balanced, less rightward-lateralized activity rather than a consistent increase in leftward asymmetry. Specifically, at low average activation, these regions exhibit strong rightward asymmetry, whereas higher activation is associated with a significant reduction in rightward bias, moving the asymmetry index closer to zero. We have clarified this distinction in the revised manuscript to avoid confusion (lines 220-229).

Comment 20: “ ‘amplitude and asymmetry appeared more modestly coupled’ – or there was no asymmetry? In other words, the coupling between asymmetry and amplitude exists when there is an asymmetry? (in which case, this is a somewhat a trivial result).”

Response: We thank the reviewer for this point. To clarify, in regions such as the medial prefrontal, superior frontal, and medial parietal cortices (including DMN areas), asymmetry is generally low. Consequently, the correlation between amplitude and asymmetry is modest, reflecting the fact that coupling is strongest in regions that exhibit robust lateralization (e.g., language and executive control regions) and naturally weaker

where there is little intrinsic asymmetry. We have revised the text to make this distinction clearer. (lines 220-229).

Comment 21: “ ‘To also examine the relationship between asymmetry and the amplitude at the network level.. ‘ – If I understood this correctly, the paragraph above reports results where the vertex-wise correlations between amplitude and asymmetry computed first and then averaged per network (it’s a bit unclear to me how p-values were computed under this procedure); the current paragraph seems to report the result when the mean amplitude and asymmetry per network are computed first, and then their association is computed. Both report some sort of averaging, “network-level”, hence it is not clearly explained what additional information is gained.”

Response: We thank the reviewer for pointing out the potential confusion. Indeed, the two analyses differ in their approach. In the vertex-wise analysis, correlations between amplitude and asymmetry are computed at each cortical vertex first, and the resulting correlation maps are then summarized per network; p-values are derived from the vertex-wise correlations using standard parametric tests and corrected for multiple comparisons. In contrast, the network-level analysis presented here first averages amplitude and asymmetry across all vertices within a network for each individual and task epoch, and then computes correlations across individuals; p-values are also derived from Pearson correlations. This complementary approach reduces spatial noise and highlights network-level effects that may be less apparent in vertex-wise maps, providing an independent perspective on amplitude–asymmetry coupling at the scale of functional networks. We have revised the text to clarify this distinction and the rationale for performing both analyses (220-229).

Comment 21: “ What do the results of this study have to say about issues raised in the introduction, such as dedifferentiation hypothesis, neurodevelopmental and psychiatric disorders?”

Response: We appreciate the reviewer’s suggestion to more explicitly connect our findings to the broader theoretical and clinical issues raised in the Introduction. Our results primarily demonstrate that mean fMRI signal amplitude is a stronger and more consistent predictor of task performance than hemispheric asymmetry, with asymmetry contributing more selectively in tasks requiring specialized processing, such as working memory and language comprehension. These observations have several implications for the dedifferentiation hypothesis, which posits that age-related or pathological reductions in hemispheric specialization may impair cognitive performance. Our findings suggest that while hemispheric asymmetry may enhance processing efficiency in specific tasks, overall neural engagement (amplitude) is a more general determinant of task performance. In other words, reduced asymmetry might be partially compensated

by stronger bilateral activation, consistent with the idea that dedifferentiation may reflect adaptive reorganization rather than purely maladaptive decline. Regarding neurodevelopmental and psychiatric disorders, many such conditions are associated with atypical lateralization or altered neural activation patterns. Our data indicate that both amplitude and asymmetry should be considered when interpreting task-related dysfunction in these populations. For example, impairments in language- or working memory-related lateralization could contribute to cognitive deficits, but reduced task engagement (lower amplitude) may have a broader impact on performance. This underscores the importance of jointly assessing both activation strength and hemispheric specialization in clinical studies to understand their relative contributions to cognitive dysfunction. We have revised the Discussion to include these points, highlighting how our results extend the relevance of amplitude and asymmetry to hypotheses about dedifferentiation, neurodevelopment, and psychiatric conditions (lines 415-423).